# HESSO: Towards Automatic Efficient and User Friendly Neural Network Training and Pruning

## Abstract

Structured pruning is a popular technique for compressing deep neural networks (DNNs) into efficient sub-networks. However, existing methods often require multi-stage process, engineering efforts, and human expertise. The Only-Train-Once series (OTOv1-v3) has been proposed to resolve some pain points by streamlining the workflow. However, the built-in sparse optimizers in the OTO series need extensive hyperparameter tuning and implicit control over sparsity, necessitating human intervention. To address these limitations, we propose the Hybrid Efficient Structured Sparse Optimizer (HESSO), which automatically and efficiently train a DNN within a single run to produce a high-performing sub-network. HESSO significantly reduces the need for manual hyperparameter tuning by solving a sparsity constrained optimization problem and enjoys user-friendly integration for generic training applications. In addition, to tackle the common issue of irreversible pruning performance collapse in certain DNNs, we further propose the Corrective Redundant Identification Cycle (CRIC), which integrates seamlessly with HESSO. The extensive numerical results showcase that HESSO can achieve competitive performance on various state-of-the-art benchmarks and support most DNN architectures. Moreover, CRIC can effectively prevent the irreversible performance collapse and further enhance the performance of HESSO on certain applications.

## 1 Introduction

Large deep neural networks (DNNs) have successfully powered a variety of applications (Goodfellow et al., 2016; Shinde & Shah, 2018; Dong et al., 2021). However, their typical significant time and space complexities make inference expensive and restrict deployment in resource-constrained environments. Consequently, how to compress the full DNN to the greatest extent while preserving the performance becomes essential in many industrial and academic AI deployment pipelines. There are various model compression techniques including but not limited to pruning (Chen et al., 2021c; 2023c; Fang et al., 2023; Wu et al., 2024), knowledge distillation (Hinton et al., 2015; Gou et al., 2021) and quantization (Han et al., 2015), which have been well developed in the past decades.

Structured pruning typically serves as the foremost technique to produce an optimal sub-network from a pre-defined full DNN by identifying and removing redundant structures (Gale et al., 2019; Han et al., 2015; Chen et al., 2021c; 2023c; Fang et al., 2023; Wang et al., 2024; Wu et al., 2024). Classical pruning methods focus on conducting a multi-stage procedure, requiring significant engineering efforts and expertise to manually build pruning search space, identify redundant structures, construct sub-network, and fine-tune to recover lost knowledge. To alleviate the human engineering burden, recent works (Chen et al., 2023c;b; Fang et al., 2023) have proposed pruning dependency graph to automate the pruning search space and sub-network construction. OTOv1-v2 (Chen et al., 2021c; 2023c) further unify these multi-stage components together, requiring only a single training run to directly get a compact sub-network without the need of further fine-tuning. Specifically, they rely on (Dual) Half-Space Stochastic Gradient Descent (D)HSPG methods to train and prune simultaneously and have introduced a rigorous theoretical version AdaHSPG+ (Dai et al., 2023).

Although OTOv1 and OTOv2 have significantly advanced the ease of use in DNN joint training and structured pruning, they still face challenges due to the complexity of the built-in (D)HSPG methods (Chen et al., 2021c; 2023c; 2020c;a). Specifically, these methods often require substantial hyper-parameter tuning for different downstream applications

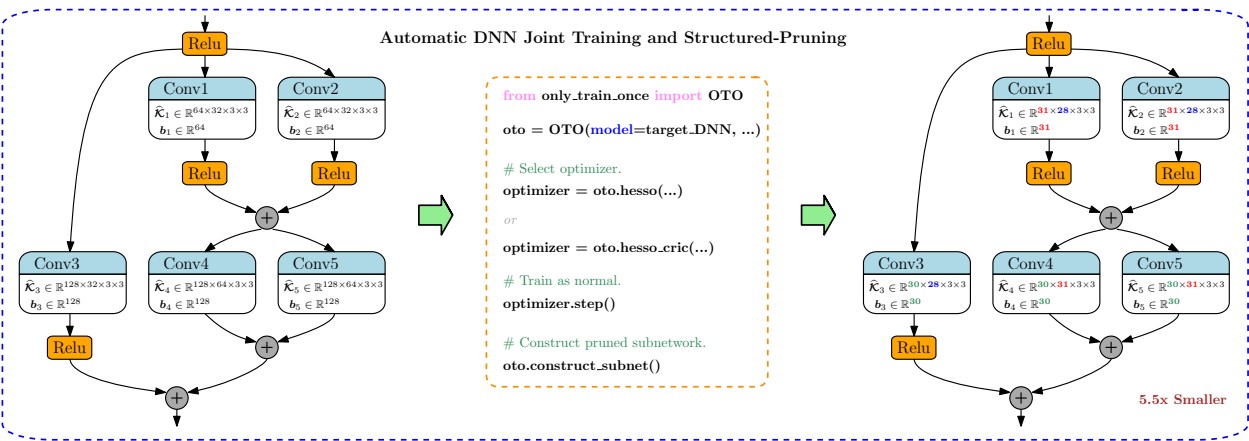

Figure 1: Automatic DNN joint training and structured pruning experience achieved by the pruning mode of OTO along with the proposed HESSO and its enhanced HESSO-CRIC optimizer. The procedure could be applied onto varying DNN and applications, and seamlessly integrated into any training pipeline to directly produce a compact pruned sub-network without further fine-tuning.

and DNN architectures (Dai et al., 2023; Wu et al., 2024). Furthermore, the sparsity explorations are implicit, which requires optimization expertise, thereby diminishes the practical convenience and usability.

Moreover, many modern pruning and neural architecture search methods rely on saliency scores (*e.g.*, Taylor based) to identify redundant structures. However, they often suffer performance degradation due to mistakenly identifying indispensable structures as redundant. This degradation can sometimes be irreversible due to architectural design constraints, transparency of training datasets, and high training resource cost, posing practical challenges for their use.

To address these issues, we propose **HESSO**: **H**ybrid **E**fficient **S**tructured **S**parse **O**ptimizer for automatic one-shot DNN training and structured pruning. Compared to the HSPG family, HESSO offers several advantages. First, it significantly simplifies the hyper-parameter setup, providing considerable practical convenience. Second, it employs a progressive pruning strategy to explicitly control the sparsity exploration, making it user-friendly. Third, HESSO optionally incorporates a novel **C**orrective **R**edundancy **I**dentification **C**ycle (**CRIC**) mechanism, which more accurately identifies redundant groups, thereby minimizing the risk of irreversible performance collapse caused by pruning indispensable structures. We now summarize our main contributions as follows.

- **Efficient Hybrid Training and Pruning Optimizer.** We propose an efficient and easy-to-use optimizer, HESSO, to enable automatic joint structured pruning and training for various model architectures and applications. HESSO progressively identifies redundant groups through flexible saliency score estimations and utilizes a hybrid training scheme to effectively transfer knowledge from redundant groups to important ones, thereby maintaining the performance of the pruned model. Compared to the D(HSPG) in OTO, HESSO explicitly controls sparsity exploration and knowledge transfer, minimizes the need for hyper-parameter tuning. As a result, HESSO becomes the first optimizer to realize convenient joint DNN training and pruning to the best of our knowledge.

- **Corrective Redundancy Identification Cycle.** We propose a novel Corrective Redundancy Identification Cycle (CRIC) to improve the accuracy of redundancy identification. CRIC addresses the approximation errors often associated with popular Taylor-based saliency scores, thereby reducing the risk of mistakenly pruning indispensable groups. CRIC employs a voting mechanism and measures the saliency scores of each group candidate using a multi-sampling approach towards the origin. CRIC can be integrated into HESSO or future joint optimizers to ensure reliable model performance by offering a more accurate assessment of group significance.

- **Numerical Experiments.** We validate the efficacy of HESSO and its enhanced version HESSO-CRIC across a variety of tasks. Specifically, we evaluate its performance on high-level computer vision tasks such as image classification and object detection, low-level vision tasks like super-resolution, as well as natural language processing tasks including large language models. The numerical results demonstrate that HESSO performs competitively, and in many cases, exceeds the state-of-the-art benchmarks, offering significant practical convenience. Additionally,

CRIC effectively mitigates the issues of irreversible collapse in pruned models, especially in challenging cases, further showcasing its utility.

## 2 Related Works

In this section, we present a brief literature review on automatic structured pruning, knowledge transfer and neural architecture optimization.

**General Pruning Procedures.** Structured pruning aims to compress DNNs by removing unnecessary structures while maintaining performance (Han et al., 2015; Wen et al., 2016). The general procedure typically involves: (*i*) training a full model; (*ii*) identifying and removing redundant structures to construct a slimmer DNN based on various criteria (Lin et al., 2019; He et al., 2018a; Wen et al., 2016; Li et al., 2020b; Zhuang et al., 2020; Chen et al., 2017; 2018; 2021a; 2020b; Gao et al., 2020; Zhuang et al., 2020; Meng et al., 2020; Yang et al., 2019; Zhou et al., 2019; van Baalen et al., 2020; Frankle & Carbin, 2018); and (*iii*) retraining the pruned model to recover any accuracy lost during pruning. These methods often require a complex and time-consuming process, involving multiple training iterations and significant domain knowledge to manually handle each step.

**Automated Pruning Given Pre-defined Search Space.** To resolve the pain points of human interventions, automated pruning is raising interests from different perspectives. Given a predefined search space, AMC (He et al., 2018b) employs reinforcement learning agents to automatically determine the optimal pruning ratio. EagleEye (Li et al., 2020a) further introduces a sub-network evaluation scheme based on adaptive batch normalization, which can be integrated into AMC. OFA (Cai et al., 2020) automates the generation of sub-networks for different hardware platforms in a single process. While these approaches yield impressive performance, their application is limited to predefined search spaces. Moreover, AMC incurs additional training costs for its reinforcement learning agent. OFA's training procedure is complex and heavy to adopt all sub-networks, which requires prior knowledge of the optimal training procedure for the largest super-network to ensure performance, making its implementation inconvenient.

**Automated Pruning Over DNNs.** On the other hand, automatically pruning arbitrary models without prior knowledge of the search space remained a significant challenge. Recent methods, such as OTO (Chen et al., 2021c; 2023c;b) and DepGraph (Fang et al., 2023), have made progress in automating the structured pruning process for general DNNs via dependency graph analysis. Subsequent works like (Wang et al., 2024) and (Ren et al., 2024) automates pruning over ONNX models. ATO (Wu et al., 2024) introduces ControlNet upon OTOv2. Among these, OTO offers a one-shot joint training and pruning framework that can seamlessly integrate into various training processes to produce high-performing sub-networks in a single run (see Figure 2 for a high-level overview). While these automated approaches have significantly improved user convenience, end-users still face significant challenges with hyper-parameter tuning and optimization expertise to calibrate OTO's built-in HSPG family (Chen et al., 2020c; Dai et al., 2023). Furthermore, some DNNs contain indispensable structures, the pruning of which leads to irreversible performance degradation. Identifying these critical structures remains an open problem that is often handled manually on a case-by-case basis, complicating practical use.

In this work, we address these challenges by proposing an efficient and user-friendly joint training and pruning optimizer, HESSO along with its enhanced version, HESSO-CRIC, which reliably identifies indispensable structures to ensure performance.

**Knowledge Transfer.** To retain the performance of a pruned sub-network, HESSO-(CRIC) incorporates a knowledge transfer mechanism through a hybrid training scheme. This approach differs from prior methods, which explicitly use knowledge distillation from unpruned models to preserve information in pruned models. Existing techniques typically require expensive computations that involve both pruned and unpruned models, either by processing logits (Lagunas et al., 2021) or the hidden activations of intermediate layers (Xia et al., 2022; Ko et al., 2023). In contrast, our approach preserves knowledge without incurring such computational costs. Another related works, ResRep (Ding et al., 2021b) and SliceGPT (Ashkboos et al., 2024), also aim to preserve computational invariance. The knowledge transfer in HESSO-(CRIC) similarly seeks to maintain computational invariance but does so by preserving objective function levels. However, SliceGPT is restricted to transformer architectures and requires manually injecting additional layers. ResRep is restricted to CNN architectures and require conducting structurally re-parametrization via

computing resetting gradients. HESSO-(CRIC) is architecture-agnostic, efficient and user-friendly, demonstrating both scalability and versatility.

**Neural Architecture Optimization.** Another related realm is the optimization over pre-specified neural architecture. NAO (Luo et al., 2018) encodes the DNN architecture into a latent representation, search over the latent space, then decodes back to a revised architecture. NAT (Guo et al., 2019) performs operator transformation upon the given DNN to produce more accurate network. These approaches transform and improve the existing DNNs, yet not search an optimal sub-network. As a result, their produced networks are typically not significantly compact compared to the baseline models. Contrarily, our approach focuses on automatically and effectively discovering compact sub-networks given pre-specified DNNs via structured pruning.

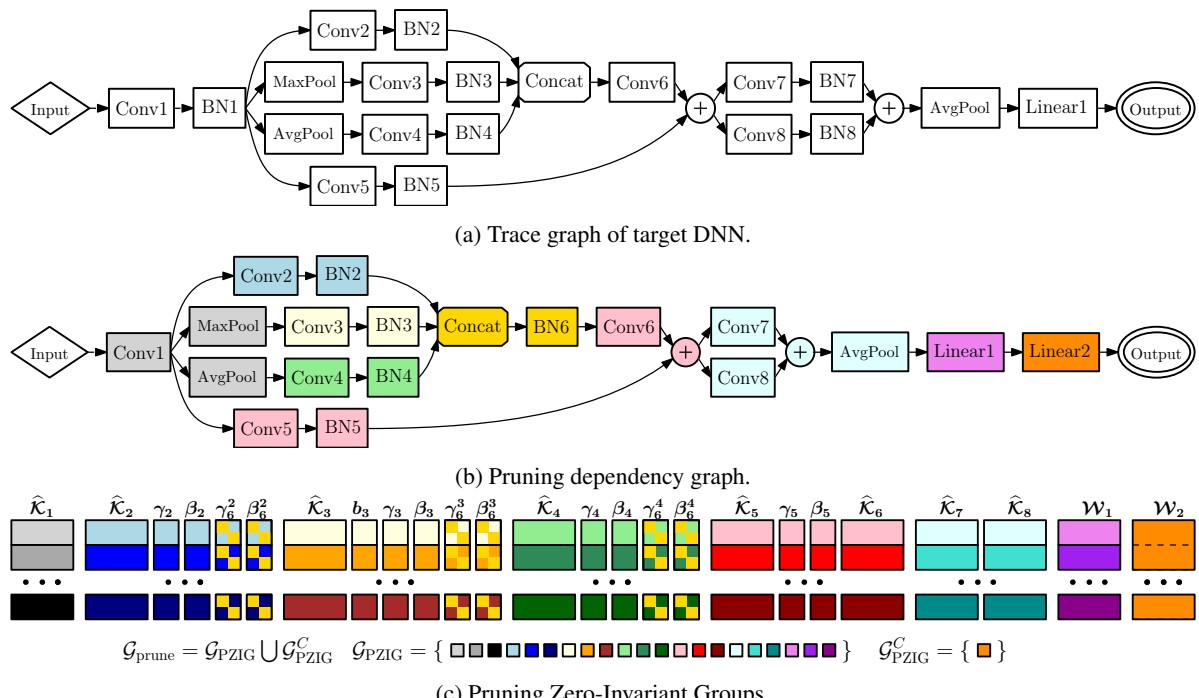

(a) Trace graph of target DNN.

(b) Pruning dependency graph.

(c) Pruning Zero-Invariant Groups.

Figure 2: Automated trainable variable partitions for one-shot structured pruning. Given the trace graph shown in Figure 2a, automatic pruning frameworks such as OTOv2 (Chen et al., 2023c) construct a pruning dependency graph shown as Figure 2b and partition the trainable variables as pruning zero-invariant groups $\mathcal{G}$ in Figure 2c.

## 3 HESSO

Given a target DNN with variables and architecture to be optimized, HESSO formulates a structured sparsity constrained optimization problem upon the set of parameter groups $\mathcal{G}$. Specifically, it aims to achieve group sparsity over the prunable variables with a target sparsity level of $K$. The optimization problem is formulated as:

$$\underset{\boldsymbol{x} \in \mathbb{R}^n}{\text{minimize}} f(\boldsymbol{x}), \quad \text{s.t. Card}\left(\{g \in \mathcal{G} | [\boldsymbol{x}]_g = 0\}\right) = K, \tag{1}$$

where $f$ is the training loss function and $[x]_g$ denotes the parameters corresponding to a parameter group $g \in \mathcal{G}$. Note that the constraint in problem 1 enforces that exactly $K$ parameter groups in $\mathcal{G}$ are pruned. One example of the set $\mathcal{G}$ is zero-invariant groups, which is introduced in (Chen et al., 2021b, Definition 1). In particular, we call $\mathcal{G}$ zero-invariant groups (ZIGs) if each group $g \in \mathcal{G}$ is zero-invariant in the sense that all of the parameters in $g$ being zeros results in its corresponding output to the next layer to be zeros as well. For details on how to identify zero-invariant groups, we refer readers to OTO framework (Chen et al., 2021b; 2023c;b) and references therein.

During the optimization process, HESSO begins with a warm-up stage, where the variables are trained using gradient descent or its variants. The purpose of the warm-up stage is to collect gradient information and guide the DNN

---

**Algorithm 1** HESSO: Hybrid Efficient Structured Sparsity Optimizer

---

1: **Input.** Initial variable $\boldsymbol{x}_0$, learning rate $\alpha$, warm-up steps $T_w$, pruning periods $P$, period length $T_p$, target group sparsity level $K$, and variable partition $\mathcal{G} = \mathcal{G}_I \bigcup \mathcal{G}_R$.

2: Warm up $T_w$ steps via SGD or its variants.

3: Initialize redundant groups $\mathcal{G}_R \leftarrow \emptyset$.

4: Initialize important groups $\mathcal{G}_I \leftarrow \mathcal{G}$.

5: Compute sparsity for each pruning period $\widehat{K} := K/P$.

6: **for** each pruning period $p = 0, 1, \cdots, P - 1$ **do**

7:     Pickup $\widehat{\mathcal{G}}_p$ in $\mathcal{G}_I$ with $\widehat{K}$-least saliency scores.

8:     Update $\mathcal{G}_I \leftarrow \mathcal{G}_I \setminus \widehat{\mathcal{G}}_p$.

9:     **for** $t = 0, 1, \cdots, T_p - 1$ **do**

10:         Compute trial iterate $\widehat{\boldsymbol{x}}_{t+1} \leftarrow \boldsymbol{x}_t - \alpha_t \nabla f(\boldsymbol{x}_t)$.

11:         Compute transferring ratio for each $g \in \widehat{\mathcal{G}}_p$:

$$[\boldsymbol{\gamma}_t]_g \leftarrow \frac{T_p - t - 1}{T_p - t} \frac{\|[\boldsymbol{x}_t]_g\|}{\|[\widehat{\boldsymbol{x}}_{t+1}]_g\|}.$$

12:         Update redundant and important variables:

$$[\boldsymbol{x}_{t+1}]_{\widehat{\mathcal{G}}_p} \leftarrow [\boldsymbol{\gamma}_t]_{\widehat{\mathcal{G}}_p} [\widehat{\boldsymbol{x}}_{t+1}]_{\widehat{\mathcal{G}}_p},$$
$$[\boldsymbol{x}_{t+1}]_{\mathcal{G}_I} \leftarrow [\widehat{\boldsymbol{x}}_{t+1}]_{\mathcal{G}_I}.$$

13:     **end for**

14:     Update $\mathcal{G}_R \leftarrow \mathcal{G}_R \cup \widehat{\mathcal{G}}_p$.

15: **end for**

16: Training important group variables till convergence.

17: **Return** the final iterate $\boldsymbol{x}^*_{\text{HESSO}}$.

---

into a relatively favorable region for convergence. Following this, HESSO enters the progressive pruning stage. At each pruning period $p$, it selects a subset $\widehat{\mathcal{G}}_p \subset \mathcal{G}_I$, consisting of groups within $\mathcal{G}_I$ that have bottom-$\widehat{K}$ saliency scores. After that, the parameters within the selected groups $\widehat{\mathcal{G}}_p$ are then gradually projected towards zero over the subsequent $T_p$ steps. Throughout the progressive pruning stage, HESSO gradually forgets the knowledge in the set $\widehat{\mathcal{G}}_p$ and conducts regular training for parameters within important groups $\mathcal{G}_I$, thereby facilitating the transfer and recapture of knowledge. We refer to this approach as hybrid training, where distinct training strategies are applied to different groups. Note that at each pruning period $p$, the parameters in the set $\mathcal{G}_R$ is not updated since they have already been projected to zero and thus removed. Therefore, we only need to update weight parameters in the set $\widehat{\mathcal{G}}_p \cup \mathcal{G}_I$. By the end of each pruning period $p$, we update the redundant groups $\mathcal{G}_R$ by merging the set $\widehat{\mathcal{G}}_p$ into $\mathcal{G}_R$. Finally, once all redundant groups are identified and are projected onto zero, the parameters in the remaining important groups $\mathcal{G}_I$ continue to be trained until final convergence. The main procedure is outlined in Algorithm 1.

**Remark 3.1.** *We claim that Algorithm 1 strictly enforces the sparsity constraint in Problem 1 to be satisfied. Specifically, during each pruning period $p$, all variables within the set $\widehat{\mathcal{G}}_p$ are projected to zero at iteration $t = T_p - 1$ since $[\gamma_t]_g = 0$ for all $g \in \widehat{\mathcal{G}}_p$ at that iteration. By the end of each pruning period, the set $\widehat{\mathcal{G}}_p$ is merged into redundant groups $\mathcal{G}_R$ and is permanently removed from further optimization. By the end of $(P-1)$th pruning period, all parameters in the redundant group $\mathcal{G}_R$ has been zeroed out. As a result, the final solution satisfies the target sparsity constraint.*

## 3.1 Saliency Score

After warming up $T_w$ steps in Algorithm 1, HESSO starts to identify redundant groups upon the target group sparsity level $K$ to partition the groups $\mathcal{G}$ into important groups $\mathcal{G}_I$ and redundant groups $\mathcal{G}_R$, i.e., $\mathcal{G}_I \bigcup \mathcal{G}_R = \mathcal{G}$ and $|\mathcal{G}_R| = K$. HESSO achieves it by periodically measuring the importance of each parameter group $g \in \mathcal{G}$. To begin, we initialize the important groups as the whole group set ($\mathcal{G}_I \leftarrow \mathcal{G}$), and the redundant groups as empty ($\mathcal{G}_R \leftarrow \emptyset$). Given a pre-defined pruning periods $P$, we identify $\widehat{K} \leftarrow K/P$ important groups to designate as redundant during each period. The redundant groups are the ones with bottom-$\widehat{K}$ saliency scores. In particular, the redundant groups $\mathcal{G}_R$ and the

important groups $\mathcal{G}_I$ are updated as follows:

$$\mathcal{G}_R \leftarrow \mathcal{G}_R \bigcup \underset{g \in \mathcal{G}_I}{\text{Bottom-}\widehat{K}} \text{ SaliencyScore}([\boldsymbol{x}]_g, [\nabla f(\boldsymbol{x})]_g),$$

$$\mathcal{G}_I \leftarrow \mathcal{G}_I \setminus \underset{g \in \mathcal{G}_I}{\text{Bottom-}\widehat{K}} \text{ SaliencyScore}([\boldsymbol{x}]_g, [\nabla f(\boldsymbol{x})]_g).$$

The selection of the saliency score in HESSO is flexible and can be tailored to different purposes. By default, we consider the categories presented in Appendix B.

## 3.2 Hybrid Training in HESSO

After identifying the redundant groups in Section 3.1, the next step involves projecting these groups onto zero and transferring their knowledge to the important groups, ensuring that the pruned model retains its performance. This is accomplished through a hybrid training scheme.

For the redundant groups $\mathcal{G}_R$, we progressively and uniformly push their parameters towards zero. This process is detailed in line 11-12 in Algorithm 1 and decipted in Figure 3. The goal is to ensure that the parameters in the redundant groups become zero after $T_p$ steps. During this penalization process, there is a risk of forgetting the knowledge contained in the redundant groups, which may manifest as a degradation in the objective function's value. To mitigate this, we employ a standard optimization method, such as vanilla SGD or its variants such as Adam, on the important groups $\mathcal{G}_I$. This step aims to continue optimizing the objective function $f$ and preserve the model's performance despite the pruning of redundant groups. By maintaining the optimization of the important groups, the knowledge lost from the redundant groups can be transferred and compensated for, ensuring that the pruned model remains effective.

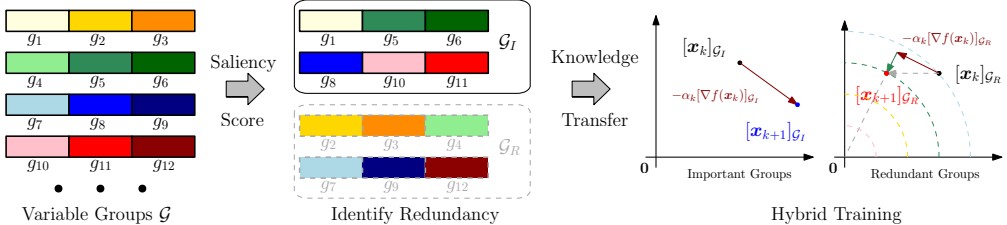

Figure 3: HESSO uses saliency scores to periodically identify redundant groups $\mathcal{G}_R$ from the group set $\mathcal{G}$ and marks the remaining groups as important groups $\mathcal{G}_I$. A knowledge transfer mechanism is proceeded by employing hybrid training strategies onto $\mathcal{G}_R$ and $\mathcal{G}_I$. In particular, the variables in $\mathcal{G}_R$ are progressively projected onto zeros after gradient descent. The important variables are kept training via gradient descent to migrate the impact of redundant project onto the objective function.

Next, we provide brief intuitive comparisons of HESSO against two popular pruning algorithms.

**Minimize tuning efforts compared to DHSPG.** DHSPG in OTOv2 involves significant hyper-parameter tuning to adjust parameters for sparsity exploration. This tuning often requires domain-specific knowledge, as the appropriate settings can vary depending on the particular application or dataset. This requirement can make DHSPG more complex and less accessible, particularly for practitioners without extensive expertise in hyper-parameter and sparse optimization. Contrarily, HESSO offers more explicit control over sparsity exploration. The pruning process in HESSO is regulated by the pruning periods $P$ and the period length $T_P$, which determine the pace and extent of the pruning procedure. This structured approach simplifies the process, making it easier to manage.

**Architecture-agnostic computational invariance compared to ResRep and SliceGPT.** ResRep (Ding et al., 2021b) and SliceGPT (Ashkboos et al., 2024) are proposed to preserve computational invariance, *i.e.*, making pruned and full models produce similar outputs, for CNNs and transformers, respectively. However, they are architecture specific, requires additional efforts, such as injecting additional layers in SliceGPT and computing reset gradients in ResRep. The knowledge transfer in HESSO similarly seeks to maintain computational invariance but does so by preserving objective function levels. In addition, HESSO is architecture-agnostic, efficient and user-friendly, demonstrating both scalability and versatility compared with ResRep and SliceGPT.

As a result, HESSO is generally easier to use and more adaptable to various applications, as it significantly reduces the need for extensive tuning and specialized knowledge. The design of hybrid training for knowledge transfer effectively promotes the performance of pruned model. It makes HESSO a more efficient and user-friendly option for achieving structured sparsity and ensuring consistent application across different tasks and domains.

## 3.3 Approximation Errors of Saliency Scores

Although HESSO can tackle most DNNs and tasks, it may yield unsatisfactory results when the target DNN possesses certain indispensable structures. We first clarify the concept of a minimal removal structure and then define when such a structure becomes indispensable. We say a structure removal if and only if the DNN without this component still serves as a valid DNN. Moreover, a removal structure is called minimal if and only if it can not be further decomposed into multiple removal structures.

**Definition 3.2** (Indispensable structure). Given a deep neural network $\mathcal{M}$, a minimal removal structure is called indispensable if removing it from $\mathcal{M}$ would cause significant performance degradation, which can not be recovered given user resources. In particular, we say a minimal removal structure as $\epsilon$-indispensable associated with an objective $f$ if pruning the variables $[\boldsymbol{x}]_g \rightarrow \boldsymbol{0}$ deteriorates $f$ at least $\epsilon$, *i.e.*, $f(\boldsymbol{x}|[\boldsymbol{x}]_g \rightarrow \boldsymbol{0}) \geq f(\boldsymbol{x}) + \epsilon$ for a minimization optimization problem. The degradation $\epsilon$ can not be recovered by *(i)* keeping training $\mathcal{M}$, *(ii)* the training cost such as GPU days exceeding user budget, or *(iii)* the training receipt for $\mathcal{M}$ is black-box and hard to be reproduced.

The origin of indispensable structures varies. One reason may be due to architectural design issues where certain layers in $\mathcal{M}$ play more critical roles than others and are very sensitive to any modifications, as exemplified by a low-level vision benchmark in Section 4.1. Another reason could be the learning strategy. For instance, in large language models (LLMs), it has been observed that knowledge is unevenly distributed across different layers (Chen et al., 2023a). Removing any of these structures could result in an irreversible collapse of the DNN's performance.

**Saliency score approximation errors.** Existing saliency scores may fail to accurately identify indispensable structures. As detailed in Appendix B, these scores are typically designed to approximate the effect of projecting groups of variables to zero on the objective function. A key limitation arises with common scores, such as Taylor-based importance scores: these scores are only reliable when the current iterate is very close to the origin. Unfortunately, this "proximity to the origin" condition is rarely met in real-world training and pruning scenarios. For instance, with Taylor-based importance scores, the approximation error grows proportionally with $\|[\boldsymbol{x}]_g\|$, as noted in Proposition 3.3. This means that the farther the iterate is from the origin, the greater the error becomes. Consequently, this can result in the false positive pruning of essential structures, leading to performance degradation.

**Proposition 3.3** (Approximation error of Taylor importance). *Suppose the gradient and second-order derivative of $f$ are bounded. Use first-order $m^L$ and second-order $m^Q$ Taylor approximations to measure the function value $f$ after pruning $g \in \mathcal{G}$, i.e., $[\boldsymbol{x}]_g \rightarrow \boldsymbol{0}$. Let $\boldsymbol{s}$ satisfy $[\boldsymbol{s}]_{\mathcal{G}\setminus g} = [\boldsymbol{0}]_{\mathcal{G}\setminus g}$ and $[\boldsymbol{s}]_g = -[\boldsymbol{x}]_g$, Then the approximation error bound $|f(\boldsymbol{x}+\boldsymbol{s}) - m^L(\boldsymbol{x}+\boldsymbol{s})|$ and $|f(\boldsymbol{x}+\boldsymbol{s}) - m^Q(\boldsymbol{x}+\boldsymbol{s})|$ are proportional to $\mathcal{O}(\|[\boldsymbol{x}]_g\|^2)$ and $\mathcal{O}(\|[\boldsymbol{x}]_g\|^3)$, respectively.*

## 3.4 Corrective Redundancy Identification Cycle

To address the limitations discussed in Section 3.3, we propose a novel Corrective Redundant Identification Cycle (CRIC). This method aims to more reliably identify redundant structures within the target DNN, even when indispensable structures are present. The CRIC mechanism can be seamlessly integrated into HESSO, enhancing its ability to accurately discern which parts of the model can be pruned without compromising performance.

To mitigate the issue of false positive redundant predictions caused by approximation errors, such as Taylor expansion, CRIC measures the saliency score of redundant group candidate multiple times along their projection towards the origin. Unlike the greedy strategy used in HESSO, CRIC incorporates a corrective cycle mechanism. This mechanism iteratively labels groups as redundant and tracks the violating groups, a group that was initially predicted as redundant (i.e., low saliency score), but later exhibits a relatively high saliency score during re-evaluation, suggesting that it may have been misclassified. To capture these inconsistencies, we utilize the violating group set $\mathcal{V}$, which collects groups that appear more redundant or deviate significantly from the current redundancy estimate. This allows CRIC to refine its decisions across iterations.

---

**Algorithm 2** Corrective Redundant Identification Cycle (CRIC)

1: **Input.** Trainable variable $\boldsymbol{x}$, learning rate $\alpha$, termination tolerance $\mathcal{T}$, target group sparsity $K$, sampling steps $T$, and prunable variable partition $\mathcal{G}$.
2: Initialize $\mathcal{S}$ to store saliency scores for each $g \in \mathcal{G}$.
3: Initialize violating group set $\mathcal{V}$:

$$\mathcal{V} \leftarrow \{g : g \in \mathcal{G} \text{ with bottom-K saliency scores}\}.$$

4: Initialize historical set $\mathcal{H} \leftarrow \mathcal{V}$.
5: **while** $|\mathcal{V}| > \mathcal{T}$ **do**
6:     Initialize trial violating group set $\widehat{\mathcal{V}} \leftarrow \emptyset$.
7:     Initialize $\alpha_0 \leftarrow \alpha$, $\lambda_0 \leftarrow \lambda$, and $\boldsymbol{x}_0 \leftarrow \boldsymbol{x}$.
8:     **for** $t = 0, 1, \cdots, T-1$ **do**
9:         Compute trial $\tilde{\boldsymbol{x}}_{t+1} \leftarrow \boldsymbol{x}_t - \alpha_t \nabla f(\boldsymbol{x}_t)$.
10:        Penalize variables in the violating set:

$$[\boldsymbol{x}_{t+1}]_{\mathcal{V}} \leftarrow \frac{T-t-1}{T-t} \frac{[\boldsymbol{x}_t]_{\mathcal{V}}}{\|[\tilde{\boldsymbol{x}}_{t+1}]_{\mathcal{V}}\|}.$$

11:        Compute saliency scores of $\mathcal{G}$ and merge to $\mathcal{S}$.
12:        Update set $\widehat{\mathcal{V}}$ if new violating groups appear:

$$\widehat{\mathcal{V}} \leftarrow \widehat{\mathcal{V}} \cup \left(\{g : g \in \mathcal{G} \text{ with bottom-K scores}\} \setminus \mathcal{V}\right).$$

13:        Update penalty $\lambda_t$ and learning rate $\alpha_t$.
14:     **end for**
15:     Update violating set $\mathcal{V} \leftarrow \widehat{\mathcal{V}} \setminus \mathcal{H}$.
16:     Update historical set $\mathcal{H} \leftarrow \mathcal{H} \bigcup \mathcal{V}$.
17: **end while**
18: Set redundant set $\mathcal{G}_R$ upon saliency score collection $\mathcal{S}$:

$$\mathcal{G}_R \leftarrow \{g : g \text{ with bottom-K scores in } \mathcal{S}\}.$$

19: **Return.** Identified redundant groups $\mathcal{G}_R$ and important groups $\mathcal{G}_I$ as $\mathcal{G} \setminus \mathcal{G}_R$.

---

As shown in Algorithm 2, $\mathcal{V}$ is initialized as the bottom-$K$ saliency score groups in line 3. A historical set $\mathcal{H}$ is used to track groups whose saliency scores have been fully exploited through multiple sampling along the projection to the origin. This set is initialized as empty set in line 4.

The corrective cycle continues as long as the violating set is large, specifically when $|\mathcal{V}| > \mathcal{T}$, where $\mathcal{T}$ is a user-defined termination threshold (defaulting to $\emptyset$). During each cycle, groups in $\mathcal{V}$ are progressively projected toward zero, with saliency scores re-evaluated at uniformly spaced points along the projection path. Groups with persistently low saliency scores and not yet visited in $\mathcal{H}$ are added to a new set $\hat{\mathcal{V}}$ for the next iteration. The cycle stops when the number of violating groups becomes negligible, i.e., when $|\mathcal{V}| \leq \mathcal{T}$. Through this corrective refinement process, CRIC effectively reduces false positive redundancy predictions and addresses failure cases of HESSO, as demonstrated in Section 4.

Theorem 3.4 guarantees that CRIC terminates within a finite number of iterations, preventing endless loops and executing efficiently. We provided detailed proof for Theorem 3.4 in Appendix A. Furthermore, Corollary 3.5 provides an upper bound on the number of cycles required by CRIC, ensuring a practical and efficient pruning process.

**Theorem 3.4** (Finite termination of CRIC). *The corrective redundancy identification cycle (Algorithm 2) terminates within a finite number of steps for any termination tolerance $\mathcal{T}$.*

**Corollary 3.5** (Upper bounds of cycle numbers). *Given the termination tolerance $\mathcal{T}$, CRIC terminates with no more than $(|\mathcal{G}| - K)/\max\{\mathcal{T}, 1\}$ cycles.*

Once the corrective cycles terminate, the saliency scores obtained are deemed reliable. At this point, the redundant set $\mathcal{G}_R$ is constructed based on these reliable saliency scores, as indicated in line 18. This set of redundant groups is then returned for further use, such as hybrid training in HESSO (as detailed in Algorithm 1). For simplicity, the HESSO variant that utilizes CRIC for identifying redundant groups is referred to as HESSO-CRIC throughout the paper, as outlined in Algorithm 3. This naming convention distinguishes the variant from HESSO, highlighting the addition of the corrective cycle mechanism that enhances the reliability of the pruning process.

## 4 Numerical Experiments

We numerically demonstrate the efficacy of HESSO across a wide range of applications, from low-level vision tasks such as super-resolution (Zhou et al., 2024), to high-level vision tasks including image classification (He et al., 2016), and object detection (Shi et al., 2020), as well as natural language processing tasks such as question answering (Rajpurkar et al., 2016) and the popular foundational large language models (Ding et al., 2023). The architectures used in these experiments encompass a variety of CNN benchmarks (Chen et al., 2023c) and transformers (Vaswani et al., 2017). These experiments involve training either from scratch or using a pre-trained checkpoint (when available) to validate the versatility of HESSO-(CRIC). Furthermore, we provided ablation studies of HESSO-(CRIC) in Section 4.6, hyper-parameters sensitivity analysis in Section 4.7, hyper-parameter tuning effort studies in Section 4.8, and computational complexity analysis in Appendix D. For details of the experiment setup, we refer readers to Appendix E.

### 4.1 Super Resolution

Table 1: Structurally pruning CARNx2.

| Optimizer | Exclusion of Dispensable Structure | Group Sparsity | # of Params | FLOPs | PSNR | | |
|---|---|---|---|---|---|---|---|
| | | | | | Set14 | B100 | Urban100 |
| Baseline | – | – | 100% | 100% | 33.5 | 32.1 | 31.5 |
| DHSPG | Manual | 50% | 24.1% | 24.3% | 33.2 | 31.9 | 31.1 |
| DHSPG | No | 50% | ✗ | ✗ | ✗ | ✗ | ✗ |
| **HESSO** | Manual | 20% | 66.8% | 66.9% | **33.5** | **32.1** | |
| **HESSO** | Manual | 30% | 50.6% | 50.8% | 32.3 | 32.0 | 31.5 |
| **HESSO** | Manual | 40% | 39.7% | 40.0% | 33.3 | 32.0 | 31.3 |
| **HESSO** | Manual | 50% | 30.5% | 30.8% | 33.2 | 31.9 | 31.1 |
| **HESSO** | Manual | 60% | | | 33.1 | 31.8 | 31.0 |
| **HESSO** | No | 50% | ✗ | ✗ | ✗ | ✗ | ✗ |
| **HESSO-CRIC** | **Automatic** | 20% | 66.8% | 66.9% | **33.5** | **32.1** | **31.8** |
| **HESSO-CRIC** | **Automatic** | 30% | 53.2% | 53.4% | 33.4 | 32.1 | 31.7 |
| **HESSO-CRIC** | **Automatic** | 40% | 40.1% | 40.4% | 33.3 | 32.0 | 31.5 |
| **HESSO-CRIC** | **Automatic** | 50% | 28.4% | 28.7% | 33.3 | 32.0 | 31.3 |
| **HESSO-CRIC** | **Automatic** | 60% | **17.7%** | **18.1%** | 33.2 | 31.9 | 31.1 |

We first selected the popular CARN architecture (Ahn et al., 2018) for the super-resolution task with a scaling factor of two, referred to as CARNx2. The benchmark DIV2K dataset (Agustsson & Timofte, 2017) was used for training, while Set14 (Zeyde et al., 2010), B100 (Martin et al., 2001), and Urban100 (Huang et al., 2015) datasets were employed for evaluation. Initially, we utilized OTO's pruning dependency analysis to identify minimal removable structures and partitioned the trainable variables into pruning-zero-invariant groups. However, directly applying DHSPG or HESSO led to significant performance degradation that was not reversible. This issue stems from the architectural design, where the penultimate convolutional layer is critical for generating satisfactory visual results, making it an indispensable structure. Pruning this layer caused the remaining filters to fail in generating reasonable visual outcomes. However, the saliency score deems them as redundant due to significant approximation errors and thus, results in irreversible performance collapse.

OTOv2 (Chen et al., 2023c) manually excluded these indispensable structures from pruning. However, this manual identification is time-consuming and requires expert knowledge. To address this, we applied HESSO-CRIC to CARN and observed that it automatically identified these crucial structures as important groups, leading to a successfully high-performing pruned model. As shown in Table 1, when manually excluding indispensable structures, both DHSPG and HESSO significantly reduced FLOPs and parameters by approximately 33% to 80%, with negligible PSNR degradation. HESSO-CRIC achieved a better trade-off between FLOPs reduction and PSNR, as demonstrated by exhibiting the frontier curve under varying pruning ratios. Visual examples shown in Figure 7 at Appendix F further cross-verify the performance preservation by our approaches.

## 4.2 Image Classification

We first employed HESSO-(CRIC) to structurally prune a pretrained OFA network (Cai et al., 2020) on the benchmark ImageNet (Deng et al., 2009). The OFA network was produced by searching from a MobileNetV3 based super-network and could achieve 80.0% top-1 test accuracy on ImageNet. We find that both HESSO-(CRIC) could effectively discover pruned sub-networks with similar size and MACs while with higher performance than other OFA networks, *i.e.*, 78.6% and 78.2% versus 76.9% test accuracy.

Table 2: Structurally pruning MobileNet Search Space.

| Method | # of Params | FLOPs | Top Acc-1 (%) |
|---|---|---|---|
| OFA$_{LARGE}$ # 75 (Cai et al., 2020) | 100% | 100% | 80.0 |
| MobileNetV2 (Sandler et al., 2018) | 37.2% | 50.4% | 72.0 |
| MobileNetV3-Large (Howard et al., 2019) | 59.1% | 36.8% | 75.2 |
| OFA # 75 (Cai et al., 2020) | 63.6% | 38.7% | 76.9 |
| **HESSO** | 61.3% | 36.9% | 78.2 |
| **HESSO-CRIC** | 62.5% | 37.8% | **78.6** |

We next benchmark ResNet50 (He et al., 2016) on ImageNet. As shown in Figure 4, HESSO-CRIC approximately forms a Pareto frontier in the trade-off between top-1 accuracy and FLOPs reduction across a range of group sparsity levels from 50% to 90%. Both HESSO and DHSPG perform competitively, although they lie within the Pareto frontier. Notably, all aforementioned three methods produce structurally pruned sub-networks that are smaller in size, require fewer FLOPs, and achieve higher accuracy compared to most existing approaches (Huang & Wang, 2018; Zhou et al., 2019; Ding et al., 2021a; Wu et al., 2024; Yang et al., 2019; You et al., 2019; Zhou et al., 2019). These results demonstrate the effectiveness of our proposed joint pruning and training optimizer on this widely used structured pruning benchmark. Finally, we note that the ATO method (Wu et al., 2024) achieves slightly higher top-1 accuracy at around 50% FLOPs reduction compared to HESSO-CRIC. This is because ATO is designed to maximize performance through a bi-level optimization framework involving a learnable mask and a task-specific Con-

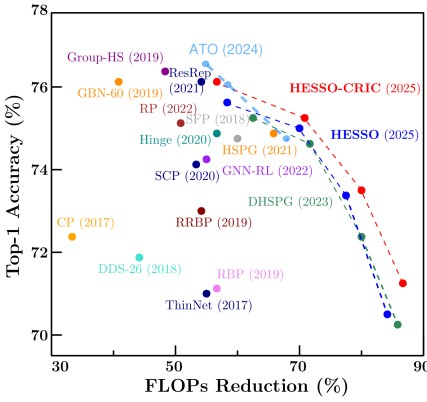

Figure 4: ResNet50 on ImageNet.

trolNet. While this enhances accuracy, it comes at the cost of increased complexity, reduced efficiency, and limited generality.

## 4.3 Object Detection

Next, we tested HESSO on the popular YOLO (Redmon et al., 2016) object detection model using the COCO benchmark dataset (Lin et al., 2014). Table 3 presents the structured pruning results for YOLOv5l (Jocher et al., 2022). Note that we selected YOLOv5l to facilitate comparisons

Table 3: Structurally pruning Yolov5l on COCO.

| Method | # of Params | FLOPs | mAP$_{0.5}$ | mAP$_{0.5:0.95}$ |
|---|---|---|---|---|
| Baseline | 100% | 100% | 66.31% | 47.71% |
| HFP (Enderich et al., 2021) | 50% | 49.5% | 63.5% | 43.4% |
| TCFP (Jeon et al., 2022) | 50% | 53.5% | 61.8% | 42.7% |
| **HESSO** (30% group sparsity) | **49%** | **52.3%** | **63.1%** | **44.4%** |
| **HESSO-CRIC** (30% group sparsity) | **49%** | **52.3%** | **63.1%** | **44.5%** |

with other existing benchmarks. We applied HESSO and HESSO-CRIC with a target group sparsity of 30%, resulting in a sub-network containing 49% of the original parameters. This allows for direct comparison with benchmarks that retain 50% of the model's parameters. The results show that a single run of HESSO and HESSO-CRIC achieved significantly higher Mean Average Precision (mAP) compared to other pruning approaches, which often require more complex, multi-stage procedures. Further visualization details can be found in Figure 7 in Appendix F.

Table 4: Structurally pruning Bert on SQuAD.

| Method | Group Sparsity | # of Params | F1-score |
|---|---|---|---|
| Baseline | 100% | 88.3% | 88.5% |
| ProxSSI (Deleu & Bengio, 2021) | – | 83.4%† | 82.0% |
| HSPG (Chen et al., 2021c) | – | 91.0% | 84.1% |
| HSPG (Chen et al., 2021c) | – | 66.7% | 82.0% |
| DHSPG | 10% | 93.3% | 87.7% |
| DHSPG | 30% | 80.1% | 87.3% |
| DHSPG | 50% | 68.3% | 86.2% |
| DHSPG | 70% | 55.0% | 83.8% |
| **HESSO** | 10% | 94.78% | 87.20% |
| **HESSO** | 30% | 84.33% | 86.72% |
| **HESSO** | 50% | 73.88% | 86.46% |
| **HESSO** | 70% | 63.34% | 85.50% |
| **HESSO** | 90% | 53.0% | 84.25% |
| **HESSO-CRIC** | 10% | 94.78% | 87.48% |
| **HESSO-CRIC** | 30% | 84.32% | 87.10% |
| **HESSO-CRIC** | 50% | 73.88% | 86.50% |
| **HESSO-CRIC** | 70% | 63.44% | 85.96% |
| **HESSO-CRIC** | 90% | 53.0% | 84.10% |

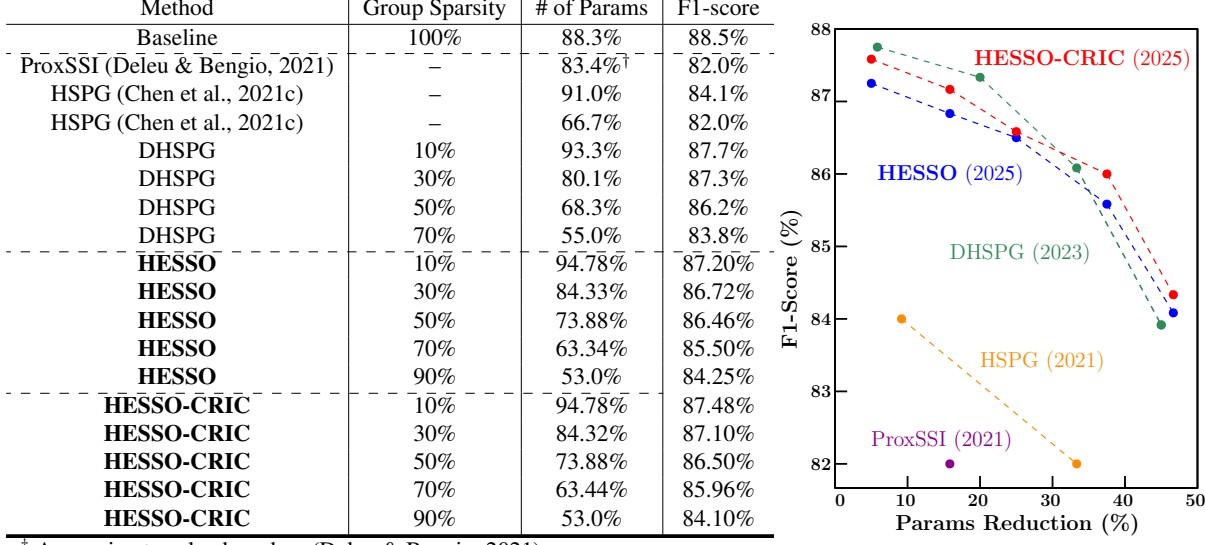

† Approximate value based on (Deleu & Bengio, 2021).

## 4.4 Question and Answering

Later, we compared HESSO-(CRIC) with DHSPG, HSPG, and a representative proximal method ProxSSI (Deleu & Bengio, 2021) for pruning a transformer model Bert (Vaswani et al., 2017), evaluated on the SQuAD question-answering benchmark (Rajpurkar et al., 2016). It is important to note that proximal methods have been standard algorithms for solving sparse optimization problems for decades. However, they are not effective at exploring sparsity while maintaining model performance in deep learning applications (Dai et al., 2023).

As shown in Table 4, HESSO, HESSO-CRIC, and DHSPG perform competitively on this task in terms of parameter reduction while maintaining F1 scores. However, DHSPG achieves these results after extensive hyper-parameter tuning, which is not convenient. HSPG penalizes all variables toward zero which severely restricts the optimization search space, leading to suboptimal performance. ProxSSI additionally lacks sufficient sparsity exploration capacity, being not comparable.

## 4.5 Large Language Model

Finally, we evaluated HESSO-(CRIC) on large language models (LLMs). Since both HESSO and HESSO-CRIC utilize full gradient information, we focused on LLMs with fewer than 3 billion parameters, such as the representative Phi-2-2.7B (Javaheripi et al., 2023), to ensure that a single 80GB GPU is sufficient, without requiring tensor parallelism (Ding et al., 2023). Our experimental setup followed that of LoRAShear (Chen et al., 2023a).

We observed that without conducting a knowledge distribution analysis and manually skipping certain layers from pruning, as LoRAShear (Chen et al., 2023a) did, HESSO often led to an irreversible performance collapse. This is because knowledge in LLMs is unevenly distributed across layers due to the learning strategy. The saliency scores calculated upon the pretraining weights may fail to identify essential structures, making it difficult to differentiate between indispensable components and those that could be pruned. As a result, pruning such critical structures severely degrades the model's performance, making recovery with limited resources nearly impossible.

HESSO-CRIC was able to automatically bypass these crucial structures, enabling effective and successful pruning. We then compared with SliceGPT (Ashkboos et al., 2024), LLM-Pruner (Ma et al., 2023), LoraShear (Chen et al., 2023a) and LoraPrune (Zhang et al., 2023) across several popular benchmarks. Our findings indicate that HESSO-CRIC consistently outperforms them at varying pruning ratios, with performance improvements becoming more pronounced as the pruning ratio increases. This is because LLM-Pruner, LoRA-Prune, and LoRAShear are LoRA-based techniques.

Table 5: HESSO-CRIC over Phi-2-2.7B.

| Pruning Ratio | Method | BoolQ | PIQA | HellaSwag | WinoGrande | ARC-e | ARC-c | OBQA | Average |
|---|---|---|---|---|---|---|---|---|---|
| Baseline | Phi-2-2.7B | 83.30 | 79.11 | 73.82 | 75.77 | 80.05 | 54.18 | 51.40 | 71.09 |
| Ratio = 20% | SliceGPT (Ashkboos et al., 2024) | 68.56 | 74.16 | 61.22 | 67.56 | 70.20 | 41.04 | 38.80 | 60.22 |
| | LLM-Pruner (Ma et al., 2023) | 61.28 | 62.79 | 36.79 | 53.12 | 52.23 | 31.06 | 30.00 | 46.75 |
| | LoraShear (Chen et al., 2023a) | 62.29 | 68.12 | 45.28 | 58.8 | 61.91 | 32.42 | 34.00 | 51.81 |
| | LoraPrune (Zhang et al., 2023) | 57.22 | 67.79 | 45.1 | 54.85 | 61.87 | 35.15 | 33.80 | 50.83 |
| | **HESSO-CRIC** | 69.67 | 74.37 | 62.27 | 66.54 | 72.30 | 41.44 | 38.20 | **60.67** |
| Ratio = 25% | SliceGPT (Ashkboos et al., 2024) | 63.70 | 71.49 | 57.72 | 66.46 | 65.86 | 38.99 | 39.80 | 57.71 |
| | LLM-Pruner (Ma et al., 2023) | 62.26 | 60.55 | 33.86 | 51.07 | 47.81 | 30.63 | 28.80 | 45.00 |
| | LoraShear (Chen et al., 2023a) | 62.17 | 64.85 | 41.27 | 55.56 | 56.52 | 30.46 | 31.80 | 48.95 |
| | LoraPrune (Zhang et al., 2023) | 62.54 | 64.69 | 40.19 | 52.33 | 56.02 | 33.62 | 32.40 | 48.83 |
| | **HESSO-CRIC** | 67.06 | 73.77 | 58.51 | 65.18 | 70.66 | 38.60 | 38.00 | **58.74** |
| Ratio = 30% | SliceGPT (Ashkboos et al., 2024) | 38.17 | 61.04 | 42.05 | 60.38 | 50.80 | 28.07 | 31.2 | 44.53 |
| | LLM-Pruner (Ma et al., 2023) | 62.11 | 59.36 | 32.27 | 51.54 | 44.07 | 30.03 | 29.8 | 44.17 |
| | LoraShear (Chen et al., 2023a) | 62.17 | 63.22 | 39.25 | 57.14 | 51.77 | 28.58 | 30.00 | 47.45 |
| | LoraPrune (Zhang et al., 2023) | 62.29 | 63.10 | 35.86 | 51.62 | 51.43 | 31.74 | 32.40 | 46.92 |
| | **HESSO-CRIC** | 67.61 | 72.14 | 53.11 | 62.75 | 62.74 | 34.81 | 36.20 | **55.62** |

Lora primarily focuses on fine-tuning well-trained models and is less effective in capturing knowledge for underfitted models, such as pruned LLMs.

## 4.6  Ablation Studies of HESSO-(CRIC)

**Hybrid VS standard training.**  To highlight the necessity of the hybrid training scheme in our HESSO design, we compare two variants: HESSO with standard training and HESSO with hybrid training. In the standard training scheme, all weight parameters in the redundant group are projected to zero in a single pruning step. In contrast, the hybrid scheme progressively project weight parameters in the redundant group to 0. As shown in Table 6, replacing hybrid training with standard training leads to a significant drop in performance across all sparsity levels. This degradation is expected, as standard training causes a substantial loss of knowledge due to the abrupt removal of parameters, whereas the hybrid scheme better preserves knowledge during pruning.

Table 6: Performance comparison between standard and hybrid training schemes under varying group sparsity levels in HESSO. Results are reported as F1 score on the BERT SQuAD dataset. The hybrid scheme consistently outperforms the standard scheme, especially at higher sparsity levels.

| | Sparsity=10% | Sparsity=30% | Sparsity=50% | Sparsity=70% | Sparsity=90% |
|---|---|---|---|---|---|
| Standard Training | 85.05% | 83.95% | 82.41% | 80.57% | 76.23% |
| Hybrid Training | 87.20% | 86.72% | 86.46% | 85.50% | 84.25% |

**HESSO over saliency scores.**  We next evaluate the performance of HESSO under various saliency score definitions. Notably, HESSO performs surprisingly poorly when using average magnitude as the saliency score. We hypothesize that this is due to the averaging effect, which compresses the score values and diminishes the contrast between important and redundant groups. This reduced discrepancy makes it more difficult to accurately distinguish and prune redundant groups, thereby increasing the likelihood of misclassification. A similar issue appears to affect the cosine similarity saliency score, particularly at high sparsity levels (e.g., 70% and 90%).

Interestingly, we also observe that one with 1st Taylor-based saliency score occasionally outperforms the one with 2rd Taylor-based saliency score. This can be explained by the fact that our approximation of the Hessian is based on the squared gradient, which does not capture true Hessian information.

Among all variants, the mixed saliency score consistently outperforms others at sparsity levels of 50%, 70%, and 90%. Moreover, HESSO with the mixed saliency score exhibits significantly lower performance variation across different sparsity levels compared to other methods. Based on these empirical findings, we adopt the mixed saliency score as the default setting (see Appendix E). A more in-depth investigation into the root causes behind the varying effectiveness of saliency scores is an important direction for future work.

Table 7: Ablation study on saliency scores for different sparsity levels. Results are reported as F1 score on the BERT SQuAD dataset.

| | Magnitude | Average Magnitude | Cosine Similarity | 1st Taylor | 2nd Taylor | Mixed |
|---|---|---|---|---|---|---|
| HESSO (30% sparsity) | 86.24% | 7.92% | 85.88% | **87.32%** | 86.87% | 86.72% |
| HESSO (50% sparsity) | 84.21% | 8.06% | 73.05% | 85.90% | 85.81% | **86.46%** |
| HESSO (70% sparsity) | 82.71% | 2.03% | 8.27% | 83.95% | 83.81% | **85.50%** |
| HESSO (90% sparsity) | 80.91% | 1.94% | 6.59% | 82.14% | 82.01% | **84.10%** |

**Mixed** = 0.2*Magnitude + 0.2*Average Magnitude + 0.2*Cosine Similarity + 0.2*1st Taylor + 0.2*2nd Taylor.

**CRIC over saliency scores.** We first conduct ablation study of CRIC under different saliency scores. The default format of CRIC primarily targets the most commonly used saliency scores that are sensitive to approximation errors caused by distances to the origin. For saliency scores with such higher sensitivities, CRIC's multiple sampling strategy—gathering information along the direction toward the origin—and its voting mechanism over historical statistics can effectively mitigate these identification issues.

To validate this, we include an ablation study for CRIC to demonstrate its improvements across varying saliency scores. As shown in the results, for commonly used saliency scores, CRIC effectively improves performance. However, magnitude and average magnitude benefits less from CRIC due to the persistence of large approximation errors, even as the groups of iterates move closer to the origin.

Table 8: Ablation Studies of CRIC on Zero-Shot Pruning Phi2.

| | Magnitude | | Avg Magnitude | | Cosine Similarity | | 1st Taylor | | 2nd Taylor | |
|---|---|---|---|---|---|---|---|---|---|---|
| | No CRIC | CRIC | No CRIC | CRIC | No CRIC | CRIC | No CRIC | CRIC | No CRIC | CRIC |
| Perplexity↓ | 629.1 | 489.4 | 713.5 | 644.6 | 525.5 | 53.4 | 438.3 | 28.6 | 378.2 | 37.1 |

Furthermore, for saliency scores whose approximation errors are not dependent on the distance to the origin, the philosophy of CRIC can still be applied with proper adaptations. In such cases, it is critical to analyze the root causes of the approximation errors for the given saliency scores. Based on these root causes, CRIC's multiple sampling strategy can be adjusted to collect more targeted signals, thereby reducing identification errors in these scenarios.

## 4.7 Hyper-parameters Sensitivity Analysis

**Pruning periods.** We first investigate the sensitivity of HESSO's performance with respect to the pruning period (see Algorithm 1). Specifically, we evaluate a range of pruning periods $\{1, 2, 5, 10\}$. As shown in Figure 5(a), the F1 score remains largely stable across these settings when the group sparsity level is fixed. A slight improvement in performance is observed as the pruning period increases. Based on this observation, we empirically choose a pruning period of 10, as recommended in our experimental setup (Appendix E).

**Sampling steps and termination tolerance.** We further investigate the sensitivity of CRIC to two key hyperparameters: the number of sampling steps and the termination tolerance (see Algorithm 2). As shown in Figure 5(b), the F1 score remains largely stable across different sampling steps $\{1, 2, 5, 10\}$ under a fixed group sparsity level. A slight performance gain is observed as the number of sampling steps increases, likely due to more opportunities to correct misclassified redundant groups. Based on this, we set the number of sampling steps to 10. Regarding termination tolerance, the F1 score shows minimal sensitivity to values $\{0, 10, 20, 50\}$, as illustrated in Figure 5(c). Increasing the termination tolerance can reduce the number of CRIC cycles (see Line 5 in Algorithm 2) and thus lower the total runtime. Accordingly, we set the termination tolerance to 50.

## 4.8 Comparative Analysis of Hyper-parameters Tuning Efforts

A key advantage of HESSO over HSPG methods in the OTO series lies in its white-box optimization design, which enables explicit control over group sparsity. In contrast to HSPGs, where achieving target group sparsity level requires extensive task-specific hyperparameter tuning (e.g., the regularization parameter $\lambda$, amplification factors $\lambda_{amplify}$, and smoothing terms $\epsilon$), HESSO only requires the specification of a desired group sparsity level. That said, no

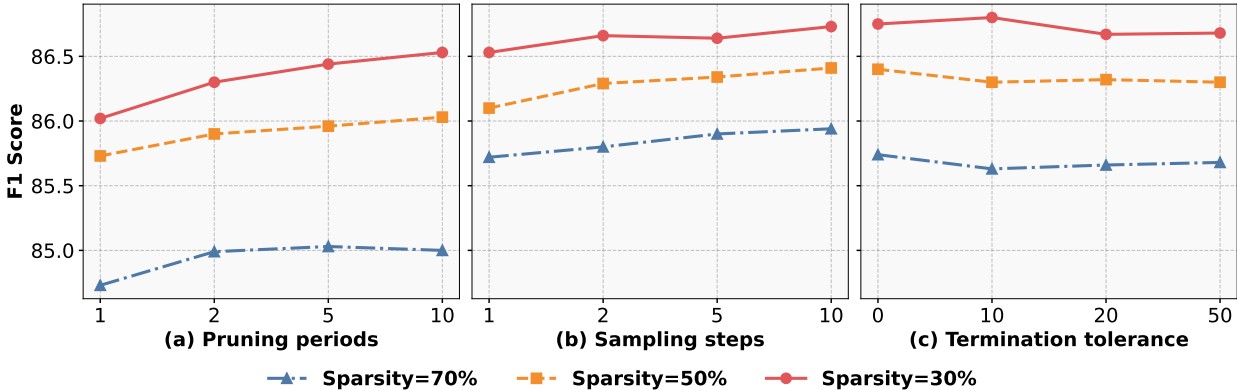

Figure 5: We evaluate the sensitivity of HESSO to three key hyperparameters: pruning periods (HESSO), sampling steps (HESSO-CRIC), and termination tolerance (HESSO-CRIC). Experiments are conducted on the BERT model using the SQuAD dataset for the Question Answering task. The F1 score is used as the evaluation metric. We consider group sparsity levels of 30%, 50%, and 70%, and report the average results over three independent runs to ensure robustness.

additional tuning of sparse-optimizer-specific hyperparameters is needed for HESSO, which significantly simplifies the training process and reduces overall tuning efforts. To illustrate this, Table 9 presents a side-by-side comparison of hyperparameter tuning requirements across several representative tasks. As shown in the **DHSPG** column, tuning HSPG requires careful sweeping over multiple hyperparameters to reach the target sparsity (e.g., sweeping over $\lambda = \{10^{-4}, 10^{-3}, 10^{-2}, 10^{-1}\}$), whereas HESSO-(CRIC) adopts a single, general-purpose recipe that applies across all tasks without modification.

Table 9: Sparse optimization related hyper-parameter tuning efforts comparisons.

|  | **HESSO-(CRIC)** | **DHSPG** |
|---|---|---|
| Super-Resolution CARNx2 | General Recipe as described in Table 12 of Appendix E. | Selected Recipe: $\lambda = 10^{-2}$, $\lambda_{amplify} = 20$, $\epsilon = 0.0$, etc. |
| Image-Classification ResNet | General Recipe as described in Table 12 of Appendix E. | Selected Recipe: $\lambda = 10^{-3}$, $\lambda_{amplify} = 2$, $\epsilon = 0.95$, etc. |
| Question-Answering Bert | General Recipe as described in Table 12 of Appendix E. | Selected Recipe: $\lambda = 10^{-3}$, $\lambda_{amplify} = 2$, $\epsilon = 0.0$, etc. |
| **Comments** | Only need to specify target group sparsity. | Grid search required over multiple hyperparameters per task to reach target sparsity. |

Additionally, this comparison focuses only on hyper-parameters specific to sparse optimizers. Black-box optimizers like HSPGs inherently manage sparsity exploration processes, which demand further tuning of broader training parameters, such as learning rate schedules and the number of epochs. In contrast, the white-box design of HESSO-(CRIC) reduces such complexities via a general recipe in Table 12 at Appendix E and low sensitivities to hyperparameters, offering a more user-friendly, efficient, and practical solution.

# 5 Conclusion

In this work, we introduced HESSO-(CRIC), a novel Hybrid Efficient Structured Sparse Optimizer tailored for pruning deep neural networks while preserving performance. By combining a hybrid training strategy with explicit, progressive pruning control, and the Corrective Redundant Identification Cycle (CRIC), HESSO-(CRIC) effectively tackles challenges such as tuning efforts, user difficulty, and irreversible performance degradation. Our experiments across diverse domains show that it not only competes with but often surpasses state-of-the-art methods.

Overall, HESSO and its enhanced version, HESSO-CRIC, represent a significant advancement in the field of structured pruning, offering a robust and versatile solution for optimizing deep neural networks with minimal human intervention. These contributions pave the way for more efficient and scalable model compression techniques, potentially leading to broader adoption in real-world applications where resource constraints are critical.

While HESSO is architecture-agnostic and compatible with full-parameter training, scaling it to larger language models requires integrating established parallelization techniques and optimization strategies (e.g., data/model/pipeline parallelism and ZeRO). As part of our future work, we plan to extend HESSO to larger LLMs such as 7B, 32B, and 70B parameter models. We are also interested in exploring its application to multimodal LLMs.

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

# A  Proof of Theorem 3.4

*Proof.* The statement is equivalent to that the violating cycle line 5-17 in Algorithm 2 terminates within finite number of steps. For convenience, we denote $\mathcal{V}_l$ as the violating set at $l$th cycle. The statement then becomes that there exists an $L < \infty$, such that $\mathcal{V}_L = \emptyset$. We now prove it as a two-step fashion.

At first, we show that the violating set at $l$th loop $\mathcal{V}_l$ is disjoint to those at all previous loops $\{\mathcal{V}_i\}_{i=0}^{i=l-1}$. This is true since the $\mathcal{V}_l$ is constructed excluding elements in the $(l-1)$th historical set $\mathcal{H}_l$

$$\mathcal{V}_l \leftarrow \widehat{\mathcal{V}}_{l-1} \setminus \mathcal{H}_{l-1}, \tag{2}$$

and $\mathcal{H}_{l-1}$ is the union of previous violating set $\mathcal{H}_{l-1} = \bigcup_{i=0}^{i=l-1} \mathcal{V}_i$. Therefore, $\mathcal{V}_l$ is disjoint to all violating sets $\{\mathcal{V}_i\}_{i=0}^{i=l-1}$.

Secondly, we prove by contradiction. Suppose there exists no an $L < \infty$, such that $\mathcal{V}_L = \emptyset$. Since $\mathcal{V}_l$ is disjoint with $\{\mathcal{V}_i\}_{i=0}^{i=l-1}$, it implies that $\mathcal{V}_l$ must include previously unseen and new element from $\mathcal{G}$. Consequently, the historical set $H_l = \bigcup_{i=0}^{i=l} \mathcal{V}_i$ will have infinite number of elements as $l$ tends to $\infty$, *i.e.*,

$$\lim_{l \to \infty} |H_l| = \infty. \tag{3}$$

However, equation 3 contradicts that the historical set $H_l$ is a subset of group partition set $\mathcal{G}$, and the cardinality of $\mathcal{G}$ is finite. Therefore, we conclude the corrective redundancy identification cycle always terminates within a finite number of steps. $\square$

# B  Saliency Score

**Magnitude.** The importance of a parameter group can be determined by its magnitude. We further normalized against all the current important instances, mapping the score into the range $[0, 1]$. Heuristically, a group of variables with lower magnitude—implying they are closer to zero—typically contributes less to the model output. Therefore, such groups are often considered less important and more likely to be pruned.

$$\begin{aligned} \text{score}_{\text{mag}}([\boldsymbol{x}]_g) &\leftarrow \|[\boldsymbol{x}]_g\|_2, \\ \text{score}_{\text{mag}}([\boldsymbol{x}]_g) &\leftarrow \frac{\text{score}_{\text{mag}}([\boldsymbol{x}]_g)}{\sum_{g \in \mathcal{G}_I} \text{score}_{\text{mag}}([\boldsymbol{x}]_g)}. \end{aligned} \tag{4}$$

**Average Magnitude.** While considering the overall magnitude can be useful, it may introduce bias by disproportionately favoring groups with more parameters, marking them as more important. To address this potential bias, the average magnitude is also considered. This metric measures the average parameter magnitude within each group, providing a normalized assessment that accounts for the number of parameters in each group. Consequently, the algorithm can more fairly compare groups of different sizes and prevent the overrepresentation of larger groups.

$$\begin{aligned} \text{score}_{\text{avg-mag}}([\boldsymbol{x}]_g) &\leftarrow \frac{\|[\boldsymbol{x}]_g\|_2}{|\sqrt{|g|}|}, \\ \text{score}_{\text{avg-mag}}([\boldsymbol{x}]_g) &\leftarrow \frac{\text{score}_{\text{avg-mag}}([\boldsymbol{x}]_g)}{\sum_{g \in \mathcal{G}_I} \text{score}_{\text{avg-mag}}([\boldsymbol{x}]_g)}. \end{aligned} \tag{5}$$

**Cosine Similarity.** Another criterion for determining group importance is the cosine similarity between the projection direction of parameter group and the negative gradient direction of the objective function. It can be calculated as the cosine similarity between $-[\boldsymbol{x}]_g$ and the negative gradient $-[\nabla f(\boldsymbol{x})]_g$, followed by a normalization to map onto a common scale. This metric evaluates whether projecting a group of parameters onto zero (*i.e.*, moving towards the origin along the negative parameter direction) aligns with a descent direction for the objective function. A descent direction is expected to decrease the objective function value, suggesting that pruning group of parameters onto zero may not significantly regress model's performance. As a result, such groups are more likely to be marked as redundant.

$$\text{score}_{\text{cosine}}([\boldsymbol{x}]_g, [\nabla f(\boldsymbol{x})]_g) \leftarrow \frac{[\boldsymbol{x}]_g^\top [\nabla f(\boldsymbol{x})]_g}{(\|[\boldsymbol{x}]_g\| \|[\nabla f(\boldsymbol{x})]_g\|)},$$
$$\text{score}_{\text{cosine}}([\boldsymbol{x}]_g, [\nabla f(\boldsymbol{x})]_g) \leftarrow \frac{\text{score}_{\text{cosine}}([\boldsymbol{x}]_g)}{\sum_{g \in \mathcal{G}_I} \text{score}_{\text{cosine}}([\boldsymbol{x}]_g)}. \tag{6}$$

**Taylor Importance.** To further quantitatively approximate the effect of projecting the parameter group $[\boldsymbol{x}]_g$ onto zero on the objective function, we can employ the Taylor expansion. Taylor expansion could estimate the impact of small changes in the parameters on the function value, allowing us to consider varying orders of Taylor importance. In particular, the first-order Taylor expansion provides a linear approximation of the objective function around the current parameter point. The impact of setting $[\boldsymbol{x}]_g \to \boldsymbol{0}$ can be estimated by the dot product of the gradient and the change in parameters. It helps identify groups whose removal likely decrease objective function.

$$\text{score}_{\text{Taylor-1st}}([\boldsymbol{x}]_g, [\nabla f(\boldsymbol{x})]_g) \leftarrow |f(\boldsymbol{x}) - f(\boldsymbol{x}|[\boldsymbol{x}]_g \to \boldsymbol{0})| \approx |[\boldsymbol{x}]_g^\top [\nabla f(\boldsymbol{x})]_g|,$$
$$\text{score}_{\text{Taylor-1st}}([\boldsymbol{x}]_g, [\nabla f(\boldsymbol{x})]_g) \leftarrow \frac{\text{score}_{\text{Taylor-1st}}([\boldsymbol{x}]_g, [\nabla f(\boldsymbol{x})]_g)}{\sum_{g \in \mathcal{G}_I} \text{score}_{\text{Taylor-1st}}([\boldsymbol{x}]_g, [\nabla f(\boldsymbol{x})]_g)}. \tag{7}$$

The second order Taylor importance is based on the second-order Taylor expansion. It includes the Hessian matrix, capturing the curvature of the objective function. This approximation considers not only the gradient but also the second derivative, providing a more accurate estimate of the impact of setting $[\boldsymbol{x}]_g \to \boldsymbol{0}$.

$$\text{score}_{\text{Taylor-2nd}}([\boldsymbol{x}]_g, [\nabla f(\boldsymbol{x})]_g) \leftarrow |f(\boldsymbol{x}) - f(\boldsymbol{x}|[\boldsymbol{x}]_g \to \boldsymbol{0})| \approx [\boldsymbol{x}]_g^\top [\nabla f(\boldsymbol{x})]_g + \frac{1}{2}[\boldsymbol{x}]_g^\top [\nabla^2 f(\boldsymbol{x})]_g [\boldsymbol{x}]_g,$$
$$\text{score}_{\text{Taylor-2nd}}([\boldsymbol{x}]_g, [\nabla f(\boldsymbol{x})]_g) \leftarrow \frac{\text{score}_{\text{Taylor-2nd}}([\boldsymbol{x}]_g, [\nabla f(\boldsymbol{x})]_g)}{\sum_{g \in \mathcal{G}_I} \text{score}_{\text{Taylor-2nd}}([\boldsymbol{x}]_g, [\nabla f(\boldsymbol{x})]_g)}. \tag{8}$$

## C   HESSO-CRIC Pseudocode

In this section, we provide the pseudocode of algorithm HESSO-CRIC.

---
**Algorithm 3** HESSO-CRIC
---
1: **Input.** trainable variable $\boldsymbol{x}_0$, learning rate $\alpha$, warm-up steps, $T_w$, and hybrid training steps $T_h$.
2: Warm-up for $T_w$ steps via SGD or its variants.
3: Use CRIC in Algorithm 2 to get redundant and important groupss $\mathcal{G}_R$ and $\mathcal{G}_I$.
4: *Hybrid Training for Knowledge Transfer.*
5: **for** $t = 0, 1, \cdots, T_h$ **do**
6:     Compute trial iterate $\widehat{\boldsymbol{x}}_{t+1} \leftarrow \boldsymbol{x}_t - \alpha_t \nabla f(\boldsymbol{x}_t)$.
7:     Compute transferring penalty ratio $[\boldsymbol{\gamma}_t]_g \leftarrow \frac{T-t-1}{T-t} \frac{\|[\boldsymbol{x}_t]_g\|}{\|[\widehat{\boldsymbol{x}}_{t+1}]_g\|}$ for each $g \in \mathcal{G}_R$.
8:     Update redundant group variables $[\boldsymbol{x}_{t+1}]_{\mathcal{G}_R} \leftarrow [\boldsymbol{\gamma}_t]_{\mathcal{G}_R}[\widehat{\boldsymbol{x}}_{t+1}]_{\mathcal{G}_R}$.
9:     Update important group variables $[\boldsymbol{x}_{t+1}]_{\mathcal{G}_I} \leftarrow [\widehat{\boldsymbol{x}}_{t+1}]_{\mathcal{G}_I}$.
10: **end for**
11: Keep training variables in important groups till convergence.
12: **Output.** The final iterate $\boldsymbol{x}^*$.
---

## D   Computational Cost Analysis

In this section, we present the time and space complexities of HESSO-(CRIC). For ease of presentation, we introduce several notations in Table 10.

Table 10: Notations.

| Symbol | Definition | Remark |
|--------|------------|--------|
| $N$ | # of trainable variables with gradient | |
| $\mathcal{G}$ | The set of parameter groups | The common setup could be pruning/erasing zero-invariant groups. |
| $|\mathcal{G}|$ | The size of $G$ | **Typically negligible compared to $N$, see the below table.** |
| $T$ | # of training steps | |
| $T_{ht}$ | # of hybrid training steps | Set as $T_{ht} = T/10$ in our generic recipe. |
| $P$ | # of pruning periods | Set as $P = 10$ in our generic recipe. |
| $S$ | # of sampling steps in CRIC | Set as $S = 10$ in our generic recipe. |
| $C$ | # of cycles in CRIC | Empirically terminates within 10 cycles. |

HESSO-(CRIC) requires additional time and space complexities while the additions are negligible. In our numerous realistic applications besides the presented academic benchmarks, HESSO-(CRIC) are quite efficient, typically as efficient as standard training via vanilla optimizers. Detailed complexity results are presented in Table 11.

Table 11: Space and Time Complexity Comparison.

| Optimizer | Variant | Space Complexity (Peak) | Time Complexity | Space Complexity Projected onto Phi2 | Time Complexity Projected onto Phi2 |
|-----------|---------|-------------------------|-----------------|--------------------------------------|--------------------------------------|
| SGD | Standard | $O(2N)$ | $O(NT)$ | $O(2N)$ | $O(NT)$ |
| **HESSO** | **SGD** | $O(2N + \|G\|)$ | $O(NT + \|G\|P)$ | $O(2.00015N)$ | $O(NT + 1.5 \times 10^{-3}N)$ |
| **HESSO-CRIC** | **SGD** | $O(2N + \|G\|S)$ | $O(NT + \|G\|P + \|G\|SC)$ | $O(2.0015N)$ | $O(NT + 1.515 \times 10^{-1}N)$ |
| Adam/AdamW | Standard | $O(3N)$ | $O(2NT)$ | – | – |
| **HESSO** | **Adam/AdamW** | $O(3N + \|G\|)$ | $O(2NT + \|G\|P)$ | $O(3.00015N)$ | $O(2NT + 1.5 \times 10^{-3}N)$ |
| **HESSO-CRIC** | **Adam/AdamW** | $O(3N + \|G\|S)$ | $O(2NT + \|G\|P + \|G\|SC)$ | $O(3.0015N)$ | $O(2NT + 1.515 \times 10^{-1}N)$ |

# E  Recommended Experimental Setup

We recommend the following hyperparameter configuration in Table 12 for HESSO and HESSO-CRIC across various applications and DNN architectures. For the target DNN to be trained and compressed, end-users likely already have a well-established training pipeline that allows the DNN to achieve high performance. To enhance usability, we recommend inheriting the hyperparameters in HESSO and HESSO-CRIC from the baseline training scheme wherever there is overlap, such as with optimizer variants and first- and second-order momentum.

Table 12: Recommended hyper-parameters and training strategies for HESSO and HESSO-CRIC.

| Hyper-parameter | Type | Recommended Setup |
|-----------------|------|-------------------|
| Optimizer variant | HESSO-(CRIC) | Inherit as the baseline optimizer. Currently support {SGD, Adam, AdamW}. |
| Group sparsity | HESSO-(CRIC) | Set upon the target pruned model size. If all variables could be pruned, the pruned model size could be approximately equal as quadratic of the density level. In addition, a randomly pruned model could be obtained by OTO's APIs. |
| First-order momentum | HESSO-(CRIC) | Inherit as the baseline optimizer's first-order momentum. |
| Second-order momentum | HESSO-(CRIC) | Inherit as the baseline optimizer's second-order momentum. |
| Weight-decay | HESSO-(CRIC) | Inherit as the baseline optimizer's weight-decay. |
| Initial learning rate | HESSO-(CRIC) | Inherit as the baseline optimizer's initial learning rate. |
| Saliency Score Criteria | HESSO-(CRIC) | By default equally considering the scores in Section 3.1. |
| Start pruning step | HESSO-(CRIC) | Set up as 1/10 of total training steps. |
| Pruning steps | HESSO-(CRIC) | Set up as 1/10 of total training steps. |
| Pruning periods | HESSO | Empirically suggest to set as 10. |
| Sampling steps | HESSO-CRIC | Empirically suggest to set as 10. |
| Termination tolerance | HESSO-CRIC | Empirically suggest to set as 50. |
| Learning rate scheduler | Training | Inherit as the baseline training, yet might need adjustments in some application to ensure the model after reaching target group sparsity is sufficiently trained under relatively large learning rate. |
| Total training steps | Training | Inherit as the baseline training and adjust upon the learning rate scheduler. |
| Start training from scratch or pre-training checkpoint | Training | Both are supported. For better performance, recommend to start from pretraining checkpoint if available. |

This inheritance strategy should also be applied to other hyperparameters related to the training pipeline, such as training steps and learning rate schedules, though some slight adjustments may be needed for some applications due

to the hybrid training process. We recommend beginning pruning at 1/10 of the total training steps and completing progressive pruning over another 1/10 of the total training steps. Because of the hybrid training stage, the learning rate schedule might require modification to ensure the DNN is sufficiently trained at a reasonably high learning rate after reaching the target group sparsity level.

Additionally, HESSO and HESSO-CRIC support training either from scratch or from a pre-trained checkpoint. For better performance and faster convergence, we recommend starting from a pre-trained status if such a checkpoint is available. We summarize the recommended hyperparameter selections and training strategies in Table 12. Note that better hyperparameter setups or training strategies may exist for specific domain tasks to achieve superior performance. For the remainder of the manuscript, we conduct experiments according to the above recommended criteria.

# F  Supplementary Pictures

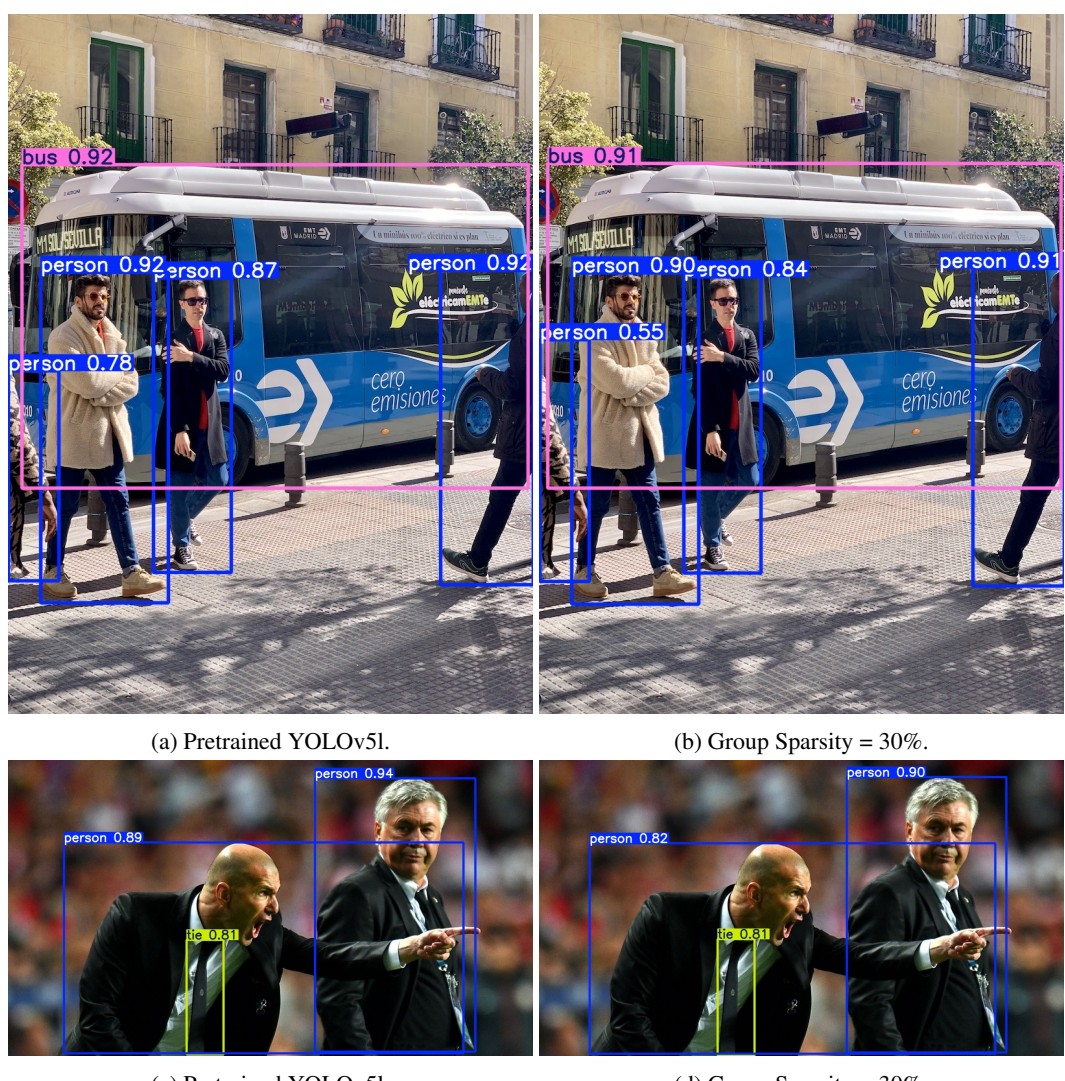

(a) Pretrained YOLOv5l.  (b) Group Sparsity = 30%.

(c) Pretrained YOLOv5l.  (d) Group Sparsity = 30%.

Figure 6: Visual examples of pruned YOLOv5l.

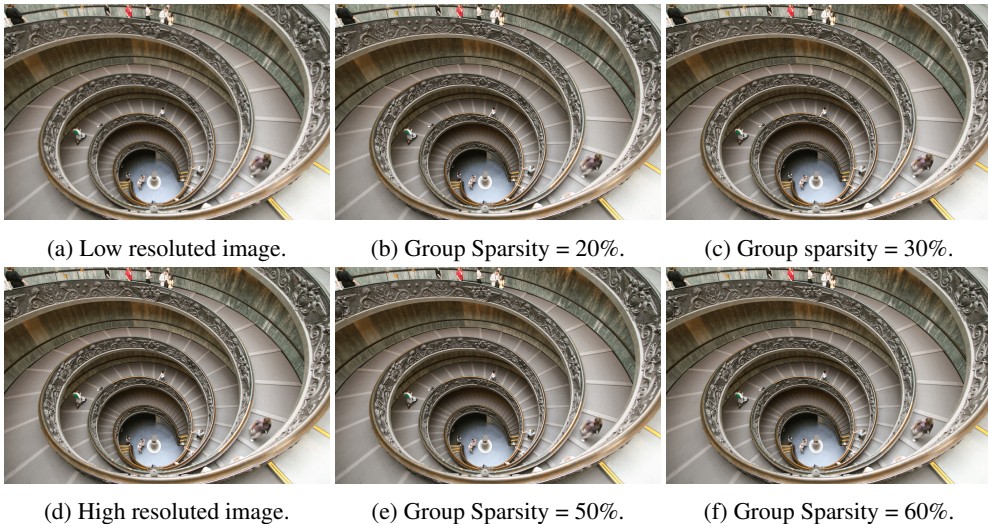

(a) Low resoluted image.      (b) Group Sparsity = 20%.      (c) Group sparsity = 30%.

(d) High resoluted image.      (e) Group Sparsity = 50%.      (f) Group Sparsity = 60%.

Figure 7: Visual examples of pruned CARNx2 produced HESSO-CRIC on Urban100.

