# OpenReview forum: "HESSO: Towards Automatic Efficient and User Friendly Neural Network Training and Pruning"
_TMLR — Rejected by TMLR_

### Review · Reviewer_LGSe · 2025-06-23

**Summary Of Contributions:**

This paper introduces the Hybrid Efficient Structured Sparse Optimizer (HESSO), which is a white-box optimizer that simultaneously trains and structurally prunes any DNN in a single run while requiring almost no task‑specific tuning. On top of that, this paper also introduces the Corrective Redundancy Identification Cycle (CRIC), which uses a multi‑sampling / voting mechanism that tries to avoid catastrophic accuracy drops when indispensable structures are mistakenly pruned. Extensive experiments over various domains have been conducted, showing that the method can achieve or exceed the accuracy–efficiency Pareto fronts of prior structured‑pruning baselines while using a single hyperparameter recipe across tasks.

**Audience:**

Yes

**Broader Impact Concerns:**

No significant ethical concerns arise directly from the presented pruning method.

**Claims And Evidence:**

Yes

**Requested Changes:**

- 1. Add ablative study isolating (a) hybrid vs standard training, (b) pruning periods P, (c) each saliency score, and (d) CRIC on/off for all domains, not only SR and LLMs.
- 2. **Report mean ± std over ≥3 runs** for key benchmarks and perform paired t‑tests versus strongest baselines.
- 3. Extend evaluation to a ≥7 B‑parameter model using gradient checkpointing or approximate layer‑wise CRIC to demonstrate scalability.
- 4. Clarify how the voting tolerance **T** and sampling steps **S** are chosen; explore sensitivity.

**Strengths And Weaknesses:**

## Strengths
- Practical usability. Only one generic hyperparameter recipe needed across tasks (vs. 3 for DHSPG), fixing the pain-point of large tuning grids for sparsity optimizers.
- Architecture agnostic. Works on CNNs, Transformers, and even LLMs without hand-crafted layer injections.
- Safety via CRIC. Prevents irreversible collapses shown on CARNx2 SR and Phi-2 pruning.
- Breadth of evaluation with competitive performance. Five domains are covered, with consistent wins.
- Theoretical clarity. CRIC’s loop-bound and termination proof give reassurance of robustness.

## Weaknesses
- Limited ablation study.  Hybrid training, progressive projection and score fusion are bundled; the paper does not show accuracy when each component is removed or replaced.
- Claim of “almost tuning‑free” is partially unsubstantiated. Only three tasks are counted in Table 7; ImageNet training still inherits the full baseline learning‑rate scheduler and epoch count, which may need adjustment when sparsity is high.
- Scalability to larger LLMs. While this work claims to work on "any DNN", the experiments stop at 2.7B parameters. Given that CRIC requires full gradients, it is unclear if memory/time overhead remains “negligible” for  7B‑ or 70B‑scale models.
- Statistical rigor. All curves are single runs; no confidence intervals or variance across seeds are reported. This is expected at least for non-LLM experiments.
- Theoretical guarantees are minimal. Theorem 3.3 only proves loop termination, not convergence to a **good** sparsity pattern or bounds on accuracy loss.

---

> ### Author Response · Authors · 2025-07-22
> **Response to Reviewer LGSe**
>
> **Q1. Limited ablation study. Hybrid training, progressive projection and score fusion are bundled; the paper does not show accuracy when each component is removed or replaced.**
>
> > A1. Thank you for your suggestion adding ablation studies. We conduct additional ablation studies using Bert on SQuad dataset to investigate the necessity of hybrid training scheme and to study how selection of saliency scores affect the HESSO's performance (See Section 4.6). As requested, we also perform the sensitivity analysis to hyperparameter pruning periods (See Section 4.7).
>
> **Q2. Claim of “almost tuning‑free” is partially unsubstantiated. Only three tasks are counted in Table 7; ImageNet training still inherits the full baseline learning‑rate scheduler and epoch count, which may need adjustment when sparsity is high.**
>
> > A2. Thank you for your constructive comments. Regarding the concern of "almost tuning‑free" claim, we refer reviewer to General Response (2). For the concern regarding ImageNet training, we gently remind the reviewer that all experiments are following the same experiment setup provided in Table 12 and the detailed results regarding ImageNet training with high sparsity is provided in Figure 4. Additionally, we have provided the source code in the supplementary materials, which can be used to reproduce and to test our results.
>
>
> **Q3. Scalability to larger LLMs. While this work claims to work on "any DNN", the experiments stop at 2.7B parameters. Given that CRIC requires full gradients, it is unclear if memory/time overhead remains “negligible” for  7B‑ or 70B‑scale models.**
>
> > A3. We appreciate your comments on LLM scalability. Please check General Response (1). Thank you.
>
> **Q4. Statistical rigor. All curves are single runs; no confidence intervals or variance across seeds are reported. This is expected at least for non-LLM experiments.**
>
> > A4. We appreciate the reviewer’s comment regarding statistical rigor. We agree that reporting variance across multiple runs is a good practice and improves reproducibility. However, we note that the competing methods that we compare with only report the single number results (i.e., the best result over multiple runs is selected). In addition, due to the computational cost of our experiments, particularly across five tasks and different sparsity levels, we prioritized reporting the best results. We believe our current findings offer meaningful comparisons and insights into the behavior of the proposed method HESSO.
>
> **Q5. Theoretical guarantees are minimal. Theorem 3.3 only proves loop termination, not convergence to a good sparsity pattern or bounds on accuracy loss.**
>
> > A5. Thank you for your comments on theoretical guarantees. We add Remark 3.1 to clarify that HESSO is guaranteed to satisfy the predetermined sparsity constraint. We hope this remark helps address your confusion on whether the final solution adheres to the desired sparsity level.
>
> **RC1. Add ablative study isolating (a) hybrid vs standard training, (b) pruning periods P, (c) each saliency score, and (d) CRIC on/off for all domains, not only SR and LLMs.**
>
> > A1. Thank you for suggested ablation studies. We refer reviewers to Section 4.6, where we provide ablation studies isolating hybrid vs standard training and each saliency score. We refer reviewer to Section 4.7 regarding ablation studies on different pruning periods, where we provide detailed sensitivity analysis.
>
> > As for the CRIC on/off ablation study, we do not understand the requested change. We gently remind the reviewer that we have provided numerical results of both HESSO and HESSO-CRIC in super resolution, image classification, object detection, and question and answering tasks. We have also provided results on how the saliency scores change with CRIC on/off using Phi-2 model. To clarify the role of CRIC, we note that it does not consistently lead to significant performance improvements across all cases. Therefore, we view CRIC primarily as a safeguard mechanism for handling challenging scenarios, such as Phi-2-2.7B model.
>
> **RC2. Report mean ± std over ≥3 runs for key benchmarks and perform paired t‑tests versus strongest baselines.**
>
> > A2. Please check responses to Q4. Thank you.
>
> **RC3. Extend evaluation to a ≥7 B‑parameter model using gradient checkpointing or approximate layer‑wise CRIC to demonstrate scalability.**
>
> > A3. Thank you for your suggestions. Please check General Response (1).
>
> **RC4. Clarify how the voting tolerance T and sampling steps S are chosen; explore sensitivity.**
>
> > A4. Thank you for your insightful comments. We refer reviewers to Section 4.7. As requested, we provide sensitivity analysis to both tolerance T and sampling steps S and specify how to choose these two parameters based on sensitivity analysis results.

---

### Review · Reviewer_qTXa · 2025-07-08

**Summary Of Contributions:**

This paper introduces HESSO, an automatic, efficient optimizer for joint training and structured pruning of neural networks. It addresses limitations in prior methods (like the OTO series) by eliminating multi-stage workflows, extensive hyperparameter tuning, and irreversible performance collapse caused by pruning critical structures. Key innovations include:

1. HESSO: A hybrid optimizer enabling single-run training/pruning via progressive group sparsity control and knowledge transfer.

2. CRIC: A corrective cycle mechanism that accurately identifies redundant groups using multi-sampling and voting, preventing irreversible collapse.

Validated across vision (image classification, super-resolution, object detection) and NLP tasks (including LLMs), HESSO can achieve competitive performance on various state-of-the-art benchmarks and support most DNN architectures.

**Audience:**

Yes

**Broader Impact Concerns:**

No concerns on the ethical implications of this work.

**Claims And Evidence:**

Yes

**Requested Changes:**

1. It seems that the authors have not uploaded the appendix. For example, it is stated that "For details of the experiment setup, we refer readers to Appendix E," perhaps I missed something, but I cannot find the appendix.

2. In Table 1, Figure 4, and Table 4, the authors labeled HESSO as (2024). However, since the paper was submitted in May 2025, the correct labeling should be HESSO (2025).

**Strengths And Weaknesses:**

Strengths:

1. This paper is well written and easy to understand.

2. HESSO enables single-run training and structured pruning for any neural network architecture, eliminating multi-stage pipelines and with little to no human intervention. Compared with previous methods (such as the OTO series), HESSO requires minimal hyperparameter tuning and can be easily integrated into existing training processes.

3. HESSO has been validated across a variety of tasks (image classification, object detection, super-resolution, natural language processing) and architectures (CNN, Transformer, LLMs such as Phi-2, etc.). Moreover, it has achieved competitive results compared with baselines.

Weaknesses:

1. There is a lack of comparison with newer baselines about automatic structure search for structured pruning. For instance, ATO [1] also proposes an end-to-end method for automatic neural network pruning during training, eliminating the need for fine-tuning. The authors should include a comparison of experimental results with ATO.

2. The authors only conducted experiments on a 2.7B LLM and lacked validation of the effectiveness of HESSO on larger LLMs (such as models with 7B parameters or more). The scalability of HESSO on larger LLMs has not been well demonstrated.

3. Termination tolerance, sampling steps, and other hyperparameters in CRIC (Algorithm 2) are set empirically without ablation. Given that the authors claim HESSO is almost tuning-free, I suggest that the authors should add ablation experiments on the hyperparameters in CRIC to demonstrate that the hyperparameters in CRIC do not significantly affect the final accuracy, thereby supporting the claim that HESSO is almost tuning-free.

4. No latency measurements. It is suggested that the authors add the inference acceleration benefits of the pruned network on CPU/GPU to further demonstrate HESSO's advantages in generating high-performance pruned networks.

[1] Wu X, Gao S, Zhang Z, et al. Auto-train-once: Controller network guided automatic network pruning from scratch[C]. Proceedings of the IEEE/CVF Conference on Computer Vision and Pattern Recognition. 2024: 16163-16173.

---

> ### Author Response · Authors · 2025-07-22
> **Response to Reviewer qTXa**
>
> **Q1. Baselines.**
>
> > A1. Thank you for your suggestion adding ATO as a competing method. We include ATO in our ResNet-50 experiments (Figure 4) and do not include it in other tasks due to ATO's complexity and unfair comparison. In particular, ATO requires a bi-level optimization framework with a task-specific ControlNet module, which must be trained separately for each task. This adds significant overhead and makes fair comparison difficult, especially for tasks beyond image classification.
>
> > For further clarification on why ATO is not a fair comparison to our methods, we provide the comparison between HESSO-(CRIC) and ATO, which is analogous to the broader trend in NAS methodologies: zero-shot NAS versus one-shot NAS.
> >    - One-shot NAS (e.g., ATO) employs bi-level differential optimization for higher performance but suffers from heavy search costs and complexity.
> >    - Zero-shot NAS (e.g., HESSO-(CRIC)) prioritizes simplicity and efficiency by designing saliency scores for fast, high-performing architecture selection, with only a slight trade-off in performance.
> > This trend is evident in recent works, where zero-shot NAS has become the predominant approach due to its practical advantages [1-5].
> >
> > [1] Lee et al., AZ-NAS: Assembling Zero-Cost Proxies for Network Architecture Search. CVPR 2024
>
> > [2] Peng et al., SWAP-NAS: Sample-Wise Activation Patterns for Ultra-fast NAS. ICLR 2024
>
> > [3] Wei et al., Auto-prox: Training-free vision transformer architecture search via automatic proxy discovery. AAAI 2024
>
> > [4] Li et al., Zero-Shot Neural Architecture Search: Challenges, Solutions, and Opportunities. TPAMI 2024
>
> > [5] Jiang et al., MeCo: zero-shot NAS with one data and single forward pass via minimum eigenvalue of correlation. NeurIPS 2024
>
> **Q2. The authors only conducted experiments on a 2.7B LLM and lacked validation of the effectiveness of HESSO on larger LLMs (such as models with 7B parameters or more). The scalability of HESSO on larger LLMs has not been well demonstrated.**
>
> > A2. Thank you for your comments. Please check General Response (1). Thanks.
>
> **Q3. Termination tolerance, sampling steps, and other hyperparameters in CRIC (Algorithm 2) are set empirically without ablation. Given that the authors claim HESSO is almost tuning-free, I suggest that the authors should add ablation experiments on the hyperparameters in CRIC to demonstrate that the hyperparameters in CRIC do not significantly affect the final accuracy, thereby supporting the claim that HESSO is almost tuning-free.**
>
> > A3. Thank you for the insightful comments. We have added sensitivity analysis of different hyperparameters in Section 4.7, which indicates that HESSO demonstrates low sensitivity to the choice of hyperparameters. For the concern regarding the claim "almost tuning-free", we refer reviewers to General Response (2).
>
>
> **Q4. No latency measurements. It is suggested that the authors add the inference acceleration benefits of the pruned network on CPU/GPU to further demonstrate HESSO's advantages in generating high-performance pruned networks.**
>
> > A4. We greatly appreciate your comments. We fully agree that wall-clock latency is an important deployment metric. However, we chose not to include it in our comparisons due to two key limitations: (1) most baselines do not report latency or release checkpoints, making direct comparisons infeasible; and (2) latency is highly dependent on hardware configurations—such as CPU vs. GPU, batch size, and runtime environment—making it difficult to report meaningful and generalizable measurements. Instead, we report FLOPs as a hardware-agnostic proxy for computational efficiency, while acknowledging that FLOPs do not perfectly correlate with real-world speed. Given these considerations, we have opted not to include wall-clock latency as an additional comparison metric.
>
>
> **RC1. It seems that the authors have not uploaded the appendix. For example, it is stated that "For details of the experiment setup, we refer readers to Appendix E," perhaps I missed something, but I cannot find the appendix.**
>
> > A1. Please check the supplementary file. The filename is HESSO_appendix.pdf. We have also included the appendix in the revised manuscript.
>
> **RC2. In Table 1, Figure 4, and Table 4, the authors labeled HESSO as (2024). However, since the paper was submitted in May 2025, the correct labeling should be HESSO (2025).**
>
> > A2. Thank you for careful revisions. We have fixed the "HESSO(2024)" typos in all figures.

---

> > ### Comment · Reviewer_qTXa · 2025-08-10
> >
> > I have reviewed the authors' rebuttal and the revised manuscript. They have done an excellent job of addressing my previous comments. All of my concerns have been resolved effectively.

---

### Review · Reviewer_XcjQ · 2025-07-10

**Summary Of Contributions:**

This paper proposes HESSO, an optimizer to replace half-space projected gradient descent (HSPG) that has been used in the OTO framework for automatic structural-pruning-and-training of neural networks. Comparing with the prior optimizer, it is claimed that the new method does not require an extensive hyperparameter tuning, and can achieve a pre-designated sparsity level more accurately. In addition, the paper also provides CRIC, which additionally refines the pruning mask for a better accuracy-to-sparsity tradeoff at the expense of using more compute.

**Audience:**

Yes

**Broader Impact Concerns:**

This paper does not need any further emphasis on the broader impact.

**Claims And Evidence:**

No

**Requested Changes:**

**Critical for acceptance**
- Please address the weaknesses above.
**Less critical**
- Surprisingly large fraction of citations seem to be placed only to increase the number of citations of the authors themselves, not to assist the readers. For instance, the first three citations---which is intended to support the point that deep neural networks are powering many applications---are all not-very-widely-used work by the same author. The same author has been cited total 6 times in the same paragraph, out of total 8 references (in fact, citing Ko et al. for distillation seems quite inappropriate). I strongly recommend citing only the works that are needed to be cited.
- The very first table, appearing in page 2, adds no academic value to the paper. In fact, it seems to be highly arbitrary, and a mere rhetoric for selling the paper, given that no concrete metric for determining the number of stars. Consider deleting the table.

**Strengths And Weaknesses:**

**Strengths.**
- A notable strength of this paper is its breadth of tasks considered. In particular, the method has been applied on models for five tasks, spanning both visual tasks (super-resolution, image classification, object detection) and language tasks (BERT for QA, Phi-2 for commonsense reasoning).
- The problem that this paper tries to address is quite important in a practical sense: reducing the need for hyperparameter tuning and hard-imposing the sparsity.

**Weakness.**
- This paper is very unclear, due to many missing definitions of key concepts. Here is an incomplete list of missing definitions, which would prevent any reader from understanding the paper. (1) f(x): The function f, first used in eq.1, has never been defined anywhere. I believe this should be the training(?) loss of the model, but need to guess this point from the context. (2) [x]_g: The bracket subscript notations appear quite frequently and plays a very important role in all algorithms, but has never been characterized properly. (3) Zero-invariant group: Only explained as "identified through pruning dependency graph analysis," which makes me believe that this is from OTOv1/v2, but this must be clearly defined in this manuscript for self-containedness, as it is not a very popular concept. (4) Outlier groups: This appears in section 3.4, and I am not sure "outlier in what sense?" (5) Violating group set: This also appears in section 3.4. (6) Minimally removal structure: I have no clue what this is, and believe that this is even grammatically wrong.
- Another thing that undermines the clarity is its many typos and mistakes. Here are some instances: (1) The paper consistently misuses \citet for the places where \citep should be used; for specific cases, see "Ko et al." and "Han et al." in the first paragraph. (2) In algorithm 1, line 5, the correct formula should be $\hat{K} := K/P$, not $T_p$. (3) The paper consistently misuses $/$ for the places where $\setminus$ (setminus) should be used. For example, see line 8 of Algorithm 1, and the second mathline of section 3.1. (4) If I am correct, lines 11 and 12 should contain some mistakes, as the weights in $\mathcal{G}_R$ are not being updated at all---only $\hat{G}_p$ is being updated. In contrast, the text says that the weights in $\mathcal{G}_R$ will be gradually updated.
- The baselines are quite limited. In many experiments, the method is only compared against the previous version, i.e., OTO with HSPG. I strongly recommend comparing the method against other competitors, e.g., ATO.
- Also, the evaluation metrics are quite random. For super-resolution, one uses FLOPS and the number of parameters. For image classification, it is the number of parameters and MAC. For object detection, it is the number of parameters only. Also, I strongly recommend comparing more "real" metrics, such as the wall-clock latency of the models.
- It is unclear whether the claimed advantages have been achieved or significant. In terms of the hyperparameter tuning, reducing three hyperparameter cases to one does not seem to be very significant. In terms of satisfying pre-determined sparsity constraint, I wish these points to be more concretely and quantitatively validated somewhere.
- Remark 3.2 is quite hand-wavy---I do not really understand the argument. It would be much clearer if a formalization of the claims have been provided.

---

> ### Author Response · Authors · 2025-07-22
> **Response to Reviewer XcjQ (Part I)**
>
> **Q1. Manuscript clarity**
>
> > A1: We sincerely thank the reviewer for the detailed and constructive feedback. We agree that clarity and completeness of definitions are critical for the readability and self-containedness of the manuscript. We have carefully revised the text to address each of the mentioned points:
>
> > (1)-(2) We acknowledge that the omission of explanations for $f(x)$ and $[x]_g$ in the manuscript. We have now clearly specified $f(x)$ and $[x]_g$ in the paragraph after problem (1).
>
> > (3) We add the explanation of zero invariant groups $\mathcal{G}$ along with references in the beginning of Section 3.
>
> > (4)-(5) We have revised Section 3.4 to provide detailed explanations for both violating group and violating group set. We omit the term outlier group in our manuscript since it has same meaning as violating group and it is not explicitly used in the algorithm description.
>
> > (6) Thank you for pointing out the grammar and clarity issues. We have revised the term to “minimal removal structure” and provided a clear definition and explanation in Section 3.4.
>
> **Q2. Typos and mistakes**
>
> > A2: Thank you for the careful reading of the paper. We would like to address each point as follows.
>
> > (1) We acknowledge the improper use of `\citet` instead of `\citep`. This has been corrected throughout the paper.
>
> > (2) We agree that the formula in Line 5 was incorrect. We have corrected it in the revised version to reflect the intended computation.
>
> > (3) The reviewer is correct that we misused the symbol `\`(backslash) where `\setminus` should have been used. This has now been consistently fixed in all relevant expressions.
>
> > (4) We appreciate the reviewer’s careful observation and would like to clarify that our notation is indeed correct.
>
> > Clarification: At pruning period $p$, the variables in the selected group $\hat{\mathcal{G}}_p$ are projected to zero by the end of the period. As a result, in the next pruning period $(p+1)$, the variables in $\mathcal{G}_R$ have already been zeroed out and do not require further updates.
>
> > Therefore, it suffices to project only the variables in $\hat{\mathcal{G}}_{p+1}$ to zero during pruning period $(p+1)$. This confirms that updating variables in $\hat{\mathcal{G}}_p$ at each pruning period is correct. To make this clear, we also reorganized the algorithm description and added extra explanations at Page 4-5.
>
> **Q3. Limited baselines**
>
> > A3. We appreciate the feedback and would like to clarify that our evaluation covers a wide range of existing approaches: comparisons with over 10 methods for ResNet50 on ImageNet, 4 methods for the MobileNet architecture, 2 methods for YOLOv5l on COCO, and 4 methods for Phi-2 2.7B. Note that the number of methods listed above exclude the HSPG and DHSPG methods. For super-resolution, we compare HSPG and DHSPG due to the lack of directly comparable methods for CARNx2.
>
> > We include ATO in our ResNet-50 experiments (See Figure 4) but exclude it from other tasks due to its complexity. In particular, ATO requires a bi-level optimization framework with a task-specific ControlNet module, which must be trained separately for each task. This adds significant overhead and makes fair comparison difficult, especially for tasks beyond image classification.
>
> > For further clarification on why ATO is not a fair comparison to our methods, we provide the comparison between HESSO-(CRIC) and ATO, which is analogous to the broader trend in NAS methodologies: zero-shot NAS versus one-shot NAS.
> >    - One-shot NAS (e.g., ATO) employs bi-level differential optimization for higher performance but suffers from heavy search costs and complexity.
> >    - Zero-shot NAS (e.g., HESSO-(CRIC)) prioritizes simplicity and efficiency by designing saliency scores for fast, high-performing architecture selection, with only a slight trade-off in performance.
> > This trend is evident in recent works, where zero-shot NAS has become the predominant approach due to its practical advantages [1-5].
>
> > [1] Lee et al., AZ-NAS: Assembling Zero-Cost Proxies for Network Architecture Search. CVPR 2024
>
> > [2] Peng et al., SWAP-NAS: Sample-Wise Activation Patterns for Ultra-fast NAS. ICLR 2024
>
> > [3] Wei et al., Auto-prox: Training-free vision transformer architecture search via automatic proxy discovery. AAAI 2024
>
> > [4] Li et al., Zero-Shot Neural Architecture Search: Challenges, Solutions, and Opportunities. TPAMI 2024
>
> > [5] Jiang et al., MeCo: zero-shot NAS with one data and single forward pass via minimum eigenvalue of correlation. NeurIPS 2024

---

> ### Author Response · Authors · 2025-07-22
> **Response to Reviewer XcjQ (Part II)**
>
> **Q4. Evaluation metric inconsistency and wall-clock latency metric**
>
> > A4. We thank the reviewer for the valuable feedback. To improve consistency, we have revised comparison tables and consistently use FLOPs as comparison metric in super-resolution, image classification and object detection tasks.
>
> > We fully agree that wall-clock latency is an important deployment metric. However, we chose not to include it in our comparisons due to two key limitations: (1) most baselines do not report latency or release checkpoints, making direct comparisons infeasible; and (2) latency is highly dependent on hardware configurations, such as CPU vs. GPU, batch size, and runtime environment, making it difficult to report meaningful and generalizable measurements. Instead, we report FLOPs as a hardware-agnostic proxy for computational efficiency, while acknowledging that FLOPs do not perfectly correlate with real-world speed. Given these considerations, we have opted not to include wall-clock latency as an additional comparison metric.
>
> **Q5. It is unclear whether the claimed advantages have been achieved or significant. In terms of the hyperparameter tuning, reducing three hyperparameter cases to one does not seem to be very significant. In terms of satisfying pre-determined sparsity constraint, I wish these points to be more concretely and quantitatively validated somewhere.**
>
> > Thank you very much for your thoughtful comments. Regarding the concern on hyperparameter tuning, we kindly refer the reviewer to our General Response (2), where we provide a more detailed discussion on this point.
>
> > As for the predetermined sparsity constraint, we have added Remark 3.1 in the revised manuscript to clarify how HESSO ensures this constraint is met. We hope this addition makes the guarantee more concrete and transparent.
>
> **Q6. Remark 3.2 clarification**
>
> > A6. Thank you for pointing this out. Our goal here was to show, using the Taylor‑based saliency score as an example, that approximation error can become significant when the current iterate is far from the origin. To make this argument clearer, we have thoroughly revised the paragraph entitled “Saliency score approximation errors” in Section 3.3.
>
> **RC1. Citations**
>
> > A1. We sincerely thank the reviewer for the thoughtful and detailed feedback. We understand the concern regarding the use of citations in the introductory paragraph and have carefully revised it to reference more widely recognized and representative works, ensuring that each citation directly supports the corresponding discussion.
>
> > In particular, we have replaced Ko et al. with the seminal work "Distilling the Knowledge in a Neural Network" (Hinton et al.) and a recent survey on knowledge distillation. We hope these changes help clarify the context and improve the manuscript’s overall clarity and credibility.
>
> **RC2. Suggestion of removing the table.**
>
> > A2. Thank you for the suggestion. As suggested, we removed the table in page 2.

---

### Author Response · Authors · 2025-07-22
**General Response**

Dear Reviewers,

> We sincerely thank all the reviewers for constructive feedback and thoughtful suggestions.

(1) We would like to first address the general concern regarding the scalability of HESSO to large language models (LLMs) with 7B parameters or more.

> First, we emphasize that HESSO is architecture-agnostic; there is no inherent limitation that prevents it from being applied to larger models. Second, HESSO operates under a full-parameter training paradigm, and scaling it to larger LLMs would primarily involve incorporating standard distributed training techniques. Specifically, data parallelism (e.g., using frameworks like DeepSpeed or Megatron-LM) would allow splitting batches across multiple GPUs; tensor/model parallelism would distribute individual layers or matrix operations across devices; and pipeline parallelism would partition layers across sequential stages of the model to maximize throughput. Additionally, memory optimization strategies like activation checkpointing, mixed-precision training (e.g., FP16 or BF16), and the Zero Redundancy Optimizer (ZeRO) are critical for reducing memory footprint and communication overhead.

> Due to the limited duration of the rebuttal period, we were unable to complete large-scale experiments on LLM >= 7B parameters. As part of our future work, we plan to integrate distributed training techniques into HESSO and evaluate its scalability on LLMs with 7B, 32B, and 70B parameters, as well as on multimodal models. This planned direction has been clarified in the conclusion section of the manuscript.

(2) Next, we would like to address the concern about the method being "almost tuning-free".

> We acknowledge that the claim is vague and can potentially confuse readers. For the sake of clarity and accuracy, we change the claim as *HESSO can significantly reduce the hyperparameter tuning by solving a sparsity constrained optimization problem compared to DHSPG and ATO* (See the changed abstract). We provide the following evidence to support this new claim.

> 1. To achieve the target group sparsity, both (D)HSPG and ATO need to perform grid search over sparse optimizer related hyperparameters including regularization parameter, amplifying factors, etc. This is due to the fact that both DHSPG and ATO need to solve a sparsity regularized optimization problem. On the other hand, one only needs to specify target group sparsity level when using HESSO. For clarity, we include the comparison over hyperparameter tuning efforts in Section 4.8.
> 2. We conduct a series of sensitivity analysis of HESSO (See Section 4.7), demonstrating that the final results are relatively insensitive to hyperparameter choices. We include pruning period, sampling steps (CRIC), and termination tolerance (CRIC) in our sensitivity analysis.
> 3. We are aware that there is no perfect quantitative evaluation metric to quantify the hyperparameter tuning efforts. To supplement this limitation, we conducted a user survey to assess the hyperparameter tuning efforts of the proposed method HESSO compared to other methods (e.g., DHSPG, ATO). We reach out to our lab collegues (A total of 10 people) and based on their experience using HESSO, we provide the following comparison table.

> **Table 1: Comparison of Different Methods**
| Method                        | Hyperparameter Tuning Effort | Extra Engineering Effort     |
|------------------------------|-------------------------------|-------------------------------|
| OTO + HESSO                  | Low                           | No                            |
| OTO + (D)HSPG                | Medium                        | No                            |
| OTO + ControlNet (ATO)       | High                          | Yes*                          |
\* Requires training a separate ControlNet module for each specific task.

---

### Decision · Action_Editor_gX8M · 2025-09-18

**Recommendation:** Reject

**Additional Comments:**

Unfortunately I believe that a minor revision would not be enough to ensure that the revised manuscript satisfied the acceptance criteria on evidence supporting the claims of the paper, and thus under the TMLR acceptance criteria I must recommend a rejection at this time. However, I would strongly encourage the authors to consider submitting a major revision with these particular issues addressed at a later time.

**Audience:**

Yes

**Audience Explanation:**

Overall the reviewers were relatively clear that there is likely an audience for the work, and the methodology, and experimental results would be valuable to some individuals in TMLR's audience. However, reviewers did also point out that the lack of evidence for some of the claims, notable for efficiency, could reduce this potential audience significantly.

**Claims And Evidence:**

No

**Claims Explanation:**

The reviewers broadly found that the authors supported many of their claims satisfactorily in the revised manuscript, with significant discussion/comments from the reviewers and authors significantly improving the manuscript. Importantly however, reviewers found a lack of evidence around two of the main claims of the work:

Efficiency: Multiple reviewers pointed out that one of the most important and fundamental claims to the work and its motivation — efficiency — is very poorly supported by the results presented, with only FLOPs reported. The authors response to the reviewers on these issues did nothing to address this particular weakness for the reviewers. Despite the work being on structured sparsity, it is not obvious that the particular structured sparsity learned by the proposed method would accelerate models on real-world hardware at least in the current writeup. Efficiency claims in training/inference are (as the author's point out) hardware dependent, and thus FLOPs does not present evidence on this front.

Hyper-parameter Tuning: The authors revised their claim that the method was "almost tuning-free" to "can significantly reduce the hyper parameter tuning..." in response to reviewer feedback during discussion. However, not all reviewers found the evidence/arguments for even the revised claim to be convincing, and instead wished to see clear quantitative evidence supporting the claim.

Given that two of the main claims of the paper were judged to be significantly lacking in convincing and clear evidence to support these claims, it's clear that as the manuscript stands, this criteria is not satisfied.

**Resubmission Of Major Revision:**

The authors may consider submitting a major revision at a later time.